# ORTHOGONAL FUNCTION REPRESENTATIONS FOR CONTINUOUS ARMED BANDITS

## ABSTRACT

This paper addresses the continuous-armed bandit problem, which is a generalization of the standard bandit problem where the action space is a $d-$dimensional hypercube $\mathcal{X} = [-1, 1]^d$ and the reward is an $s-$times differentiable function $f : \mathcal{X} \to \mathbb{R}$. Traditionally, this problem is solved by assuming an implicit feature representation in a Reproducing Kernel Hilbert Space (RKHS), where the objective function is linear in this transformation of $\mathcal{X}$. In addition to this additional intake, this comes at the cost of overwhelming computational complexity. In contrast, we propose an explicit representation using an orthogonal feature map (Fourier, Legendre) to reduce the problem to a linear bandit with misspecification. As a result, we develop two algorithms `OB-LinUCB` and `OB-PE`, achieving state-of-the-art performance in terms of regret and computational complexity.

## 1 INTRODUCTION

This paper considers the problem of optimizing a reward function $f : \mathcal{X} \to \mathbb{R}$. As in most of the literature, we will always take $\mathcal{X} = [-1, 1]^d$, as the results can be trivially extended to any compact domain with Lipschitz boundary. At each round, the learner chooses an action $x_t \in \mathcal{X}$ and observes a noisy sample for $f(x_t)$. The goal is to minimize the cumulative regret, defined as $\sum_{t=1}^{T} f(x_t) - \sup_{x \in \mathcal{X}} f(x)$, where $T$ is a given time horizon. This is also known as the continuous armed bandit problem, which generalizes, increasing the number of arms to an uncountable set, the finite "multi-armed bandit problem" (Lattimore & Szepesvári, 2020; Auer et al., 2002).

In the continuous bandit problem (Agrawal, 1995), the algorithm cannot try all the arms even once, so it has to exploit some notion of smoothness of the function $f$ to estimate the mean reward of the majority of arms without pulling them. Without any assumption of smoothness, it can be demonstrated that the problem is non-learnable. In the literature on continuous bandits, two primary families of methods have been introduced, depending on the specific assumptions made about the function $f$.

1. Lipschitzness: under the assumption that $f$ is Lipschitz or Hölder continuous, algorithms based on discretization (Kleinberg, 2004; Kleinberg et al., 2008) are able to achieve the best possible regret bound.

2. RKHS representation: under the assumption that $f$ belongs to a reproducing kernel Hilbert space with a known kernel, it is possible to solve the problem with kernel methods, and in particular Gaussian processes (Srinivas et al., 2009; de Freitas et al., 2012; Valko et al., 2013; Chowdhury & Gopalan, 2017; Shekhar & Javidi, 2018; Li & Scarlett, 2022).

The idea of the second family of algorithms, which are considered to be state of the art for practical applications, is that there is a feature map $\boldsymbol{\varphi} : [a, b] \to \mathcal{H}$, where $\mathcal{H}$ is a Hilbert space, such that for $x, x' \in [a, b]$ the kernel $k(x, x') = \langle \boldsymbol{\varphi}(x), \boldsymbol{\varphi}(x') \rangle_{\mathcal{H}}$ and an unknown vector $\boldsymbol{v}$ such that $f(x) = \langle \boldsymbol{\varphi}(x), \boldsymbol{v} \rangle_{\mathcal{H}}$. In this way, the problem is reduced to a linear bandit (Abbasi-Yadkori et al., 2011) and, by using the kernel trick (Schölkopf, 2000), it is possible to solve it even without an explicit calculation of $\boldsymbol{\varphi}$. While very elegant, this solution leads to relevant downsides: a rather specific assumption on $f$ and, most importantly, a terrible computational inefficiency since kernel methods are well-known to be very slow in prediction.

In this paper, we propose a different approach to reduce the continuous bandit problem to a linear bandit. We drop the assumption that this feature map is exact, thus admitting to have $f(x) =$

$\langle \boldsymbol{\varphi}(x), \boldsymbol{v} \rangle_{\mathcal{H}} + \varepsilon(x)$ for some error function $\varepsilon$. In this way, we are able to explicitly compute the approximate feature map by means of an orthogonal basis of a Hilbert space and reduce the problem to a misspecified linear bandit (Lattimore et al., 2020; Ghosh et al., 2017).

## 1.1 RELATED WORKS

As mentioned earlier, the main comparisons for the algorithms of this paper are kernel methods for the continuous bandit problem, which are able to solve it by viewing it as a linear bandit through an implicit representation given by a feature map in an RKHS. Thanks to their excellent performance in practical scenarios, much research has been done in this direction, starting with (Srinivas et al., 2009), which introduced the popular `GP-UCB` algorithm to (Chowdhury & Gopalan, 2017), which invented its improved version `IGP-UCB`. Improved regret bounds were obtained by (Valko et al., 2013; Li & Scarlett, 2022), while the problem of reducing the overwhelming computational complexity has been tackled in (Calandriello et al., 2019; 2020; 2022). The connection with recent advancements in this field is very deep and will be explained in detail in the appendix E.2. Recently, an idea similar to the one in our paper has been studied by Liu et al. (2021). Their method requires that the objective function $f$ is smooth in the Hölder sense, i.e., it admits continuous derivatives up to a certain order, and the last derivative considered satisfies an Hölder inequality. Their algorithm relies on both discretization of the state space, dividing the domain of $f$ in a number of bins depending on $T$, and linear bandits, making a *local* linear representation of $f$ in each bin. This is done by Taylor polynomials: function $f$ is locally approximated by its Taylor expansion. Conversely, our paper focuses on a *global* representation of the function $f$ by means of a generalized Fourier series. In this way, we obtain an algorithm that is arguably much simpler and avoids fitting a number of linear bandit instances that diverges with $T$ (thus obtaining also a better computational efficiency).

## 2 PRELIMINARIES

Let us start by introducing the main definitions and notations used in this work, which will also be summarized in a table in the appendix A. In this paper, we study the continuous armed bandit problem over the space $\mathcal{X} = [-1, 1]^d$ where the objective function $f$ is assumed to be smooth of some order $s$ known to the learner (including the case $s = +\infty$). Precisely, given that $f \in \mathcal{C}^s(\mathcal{X})$, the learner has to choose at each time step an arm $x_t \in \mathcal{X}$, receiving a reward $r_t = f(x_t) + \eta_t$, where $\{\eta_t\}_t$ are i.i.d. samples form a $\sigma-$subgaussian distribution. The goal is to minimize the regret up to a known time horizon $T$, defined as $R_T := \sum_{t=1}^{T} \sup_{x \in \mathcal{X}} f(x) - f(x_t)$. In addition to the space of $s-$ times differentiable functions $\mathcal{C}^s(\mathcal{X})$, we will make use of the function space of square-integrable functions $L^2(\mathcal{X})$, defined as

$$L^2(\mathcal{X}) := \left\{ f : \int_{\mathcal{X}} f(x)^2 \, dx < +\infty \right\}.$$

This space (if we identify functions that are equal almost everywhere) is a Hilbert space endowed with the following scalar product $\langle f, g \rangle_{L^2} := \int_{\mathcal{X}} f(x)g(x) \, dx$. In a Hilbert space, two functions are said to be orthogonal if their scalar product is zero. A set of vectors in a Hilbert space is called an *orthogonal* basis if every element of the space can be written as a linear combination of elements in the set and all the elements are pairwise orthogonal. We will see some examples of orthogonal bases in the following subsection.

## 3 ORTHOGONAL FUNCTIONS AND THEIR PROPERTIES

For simplicity of notation, we start from the case where $d = 1$, so that $\mathcal{X} = [-1, 1]$. The generalization for higher $d$ will come by making a Cartesian product of the basis functions. The most famous basis of $L^2(\mathcal{X})$ is indeed the Fourier basis. We use this basis to define a feature map associating to each $x \in \mathcal{X}$, the application of the basis to that point.

**Definition 1** (Fourier feature map). *Define, for any $n \geq 0$, the following functions*

$$\varphi_{F,n}(x) := \begin{cases} 1/\sqrt{2} & n = 0 \\ \cos\left(\frac{n}{2}\pi x\right) & n > 0 \text{ even} \\ \sin\left(\frac{n+1}{2}\pi x\right) & n \text{ odd} \end{cases} .$$

*Furthermore, for every $N \in \mathbb{N}$, define the following feature maps $\boldsymbol{\varphi}_{F,N} : [-1,1] \to \mathbb{R}^N$*

$$\boldsymbol{\varphi}_{F,N}(x) := [\varphi_{F,0}(x), \varphi_{F,1}(x) \ldots \varphi_{F,N}(x)]$$

As anticipated, the importance of the Fourier feature map lies in the fact that the set $\{\varphi_{F,n}\}_{n \in \mathbb{N}}$ forms an *orthogonal* basis for the space $L^2(\mathcal{X})$. However, the Fourier basis is not the only well-known function sequence to enjoy this property. Indeed, there are also sequences of polynomials forming an orthogonal basis of $L^2(\mathcal{X})$, the most famous one being the Legendre polynomials.

**Definition 2** (Legendre feature map). *(Quarteroni et al., 2010) Calling $\varphi_{L,n}(x)$ the n-th order Legendre polynomial, define, for every $N \in \mathbb{N}$, the following feature map $\boldsymbol{\varphi}_{L,N} : [-1,1] \to \mathbb{R}^N$*

$$\boldsymbol{\varphi}_{L,N}(x) := [\varphi_{L,0}(x), \ldots \varphi_{L,N}(x)]$$

Legendre polynomials are currently used in numerical mathematical applications like polynomial interpolation and numerical quadrature.

### 3.1 MULTI-DIMENSIONAL GENERALIZATION

Even if these basis functions are all defined on the interval $[-1,1]$, we can generalize them to the case where $\mathcal{X} = [-1,1]^d$ by doing a Cartesian product operation. Precisely, the generalization of a given basis $\boldsymbol{\varphi}_N$ of $L^2([-1,1])$ to $[-1,1]^d$ is given by

$$\boldsymbol{\varphi}_N^d(x_1, \ldots x_d) := \left\{ \varphi_{N_1}(x_1) \times \varphi_{N_2}(x_2) \ldots \varphi_{N_d}(x_d) : \sum_{i=1}^d N_i \leq N \right\}.$$

This formula applies to Fourier and Legendre bases in the same way. Unlike the 1-dimensional case, where we needed exactly $N$ features to get a basis of degree $N$, this number is significantly bigger here. In fact, it can be proved that the length of the feature vector $\boldsymbol{\varphi}_N^d(x_1, \ldots x_d)$ corresponds to $\binom{N+d}{N}$, which is always bounded by $N^d$. It is much more difficult to visualize this feature map, which goes $\mathcal{X} \to \mathbb{R}^{\binom{N+d}{N}}$ still, the mathematically we are following the very same idea of $d = 1$.

We can use these feature maps to project any function in $L^2(\mathcal{X})$ on the linear subspace generated by the first $N$ elements of the bases. Formally, being $\{\varphi_n\}_{n \in \mathbb{N}}$ the Legendre or Fourier basis functions, for any $f \in L^2(\mathcal{X})$ there are coefficients $\{a_n\}_{n \in \mathbb{N}}$ such that $\sum_{n=0}^N a_n \varphi_n \xrightarrow{L^2} f$, and they can be found with a simple scalar product: $a_n = \langle f, \varphi_n \rangle_{L^2}$. The existence of this representation is sufficient to ensure that the function $f$ can be approximated by a linear function in a features space given by our choice of $\{\varphi_n\}_{n \in \mathbb{N}}$. Not only the existence of the sequence $a_n$ is ensured, but it has remarkable properties if the function $f$ is smooth. If $f \in \mathcal{C}^s(\mathcal{X})$, as in our assumptions, the coefficients $a_n$ form a fastly decaying sequence, and the precise way in which $a_n \to 0$ depends on $s$. These kinds of results are known in Approximation Theory as decay properties.

### 3.2 DECAY PROPERTIES

The smoothness of $f$ can heavily influence the magnitude of its coefficients $a_n$ in the Fourier of Legendre basis. To give the idea behind our method, we list here two informal theorems in case of $d = 1$, that is, for $\mathcal{X} = [-1,1]$. These decay properties allow us to prove that smooth functions can be very well approximated by orthogonal polynomials or Fourier series.

**Theorem 1.** *(Informal) Let $f : [-1,1] \to \mathbb{R}$ be a measurable function. If $f \in \mathcal{C}^s([-1,1])$, and $f^{(s+1)}$ is square-integrable then,*

$$\left\| \sum_{n=0}^N a_{L,n} \varphi_{L,n} - f \right\|_\infty = \mathcal{O}(N^{-s-1/2}),$$

*The same result holds for the Fourier basis $\{\varphi_{F,n}\}_n$ if $f \in \mathcal{C}^s_{per}([-1,1])$[1].*

---

[1]Periodic conditions at the boundary.

---

**Algorithm 1** OB-**LinBand** Algorithm

---

**Require:** Linear bandit algorithm **LinBand**, Time horizon $T$, Degree $N$ of the feature map, Error probability $\delta$, Basis function to use $\{\varphi_n\}_n$, Upper bound $S$ for $\|f\|_{L^2}$
1: $\boldsymbol{\varphi}_N \leftarrow [\varphi_0(x), \varphi_1(x) \ldots \varphi_{\widetilde{N}}(x)] \; \forall x \in \mathcal{X}$
2: Instanciate learner $\mathcal{L} \leftarrow$ **LinBand**(arms$= \boldsymbol{\varphi}_N, S = S, \delta = \delta$)
3: **for** $t \in \text{Range}(T)$ **do**
4:      $x_t \leftarrow \mathcal{L}$.select arm()
5:      Receive reward $r_t$
6:      $\mathcal{L}$.update($r_t$)
7: **end for**

---

**Theorem 2.** *(Informal) Let $f : [-1, 1] \to \mathbb{R}$ be a measurable function. If $f$ is analytic, then,*

$$\left\| \sum_{n=0}^{N} a_{L,n} \varphi_{L,n} - f \right\|_{\infty} = \mathcal{O}(N^{\beta} \rho^{\alpha - N}),$$

*for some $\alpha, \beta > 0$ and $\rho > 1$. The same result holds for the Fourier basis if $f \in \mathcal{C}_{per}^s([-1, 1])$.*

For the formal statement of these results, see the appendix B.[2] Note that all the results of this section are valid in case $d = 1$. For higher dimensional spaces, we need a different strategy, as no result exists in the literature concerning orthogonal polynomials in more than one dimension. We discuss this technical difficulty in the proof of the main theorems.

## 4   ALGORITHMS

This section presents our algorithm designed to address the continuous armed bandit setting with smooth objective functions. Our algorithm is presented in two variations, each with a different focus. The first variation is tailored for practical performance, while the second variation emphasizes theoretical guarantees.

The idea of both algorithms is to use the feature maps defined in the previous section to convert our continuous bandit problem into a linear bandit one. Precisely, consider the objective of the bandit algorithm, which is to play arms with high reward to minimize the regret. We have

$$\arg\max_{x \in \mathcal{X}} f(x) \approx \arg\max_{x \in \mathcal{X}} \sum_{n=0}^{N} a_n \varphi_n(x) = \arg\max_{x \in \mathcal{X}} \langle \mathbf{a}_N, \boldsymbol{\varphi}_N(x) \rangle, \tag{1}$$

where $\mathbf{a}_N$ stands for $[a_0, \ldots a_N]$, the vector of the first $N$ coefficients of the projection of the function in the vector space generated by the first $N$ basis functions. In fact, what we do in our abstract algorithm 1 is exactly (line 1) to apply a given feature map $\boldsymbol{\varphi}_N$ on $\mathcal{X}$ and then use a generic linear bandit algorithm **LinBand** to choose the actions. The next subsections are devoted to choosing which linear bandit algorithm to use. A first solution will be to use the celebrated `LinUCB` (defined in (Abbasi-Yadkori et al., 2011) as OFUL, the name `LinUCB` was given later (Lattimore & Szepesvári, 2020) chapter 19). This will originate an elegant and practical algorithm, `OB-LinUCB`. To achieve the optimal regret guarantee, we will use a linear bandit algorithm called `phased elimination` Lattimore et al. (2020) falling in the field of *misspecificated linear bandits* (Lattimore et al., 2020; Ghosh et al., 2017). This will originate another version of our algorithm that we call `OB-PE` (see subsection 4.2).

### 4.1   OB-LINUCB

If, in equation 1 we had a perfect equality instead of the "$\approx$" symbol, the problem would be solved by standard bandit algorithms such as `LinUCB` (defined in (Abbasi-Yadkori et al., 2011) as OFUL, the name `LinUCB` was given later (Lattimore & Szepesvári, 2020) chapter 19), taking as hidden

---

[2]For the theory behind these results, refer to (Wang & Xiang, 2012; Butzer et al., 1977; Katznelson, 2004).

vector $\mathbf{a}_N$, and as space of arms the subset of $\mathbb{R}^N$ given by $\{\boldsymbol{\varphi}_N(x) : x \in \mathcal{X}\}$. Instead, in this case, we have an approximation error since we are truncating the sum to $N$, thus falling in the field of *misspecificated linear bandits* (Lattimore et al., 2020; Ghosh et al., 2017). Nonetheless, if the misspecification is very small, solving the problem using LinUCB rather than a specific algorithm for misspecified linear bandits is still more convenient.

For clarity, we call this algorithm, obtained by plugging `LinUCB` in line 1, `OB-LinUCB`. The parameter $\widetilde{N}$, which indicates the dimension of both the unknown vector $\mathbf{a}_N$ and of the feature map $\boldsymbol{\varphi}_N$, is strictly liked with $N$, which corresponds to the *degree* of the feature map. If $d = 1$, the two numbers coincide, while in the multidimensional case, their relation is described in section 3.1. Note that `LinUCB` requires an upper bound $S$ on the two norm of $\mathbf{a}_N$. Having assumed $\{\varphi_n\}_n$ to be an orthogonal sequence in $L^2$ reveals crucial in this case. Indeed, from Parseval's theorem (Rudin, 1974)

$$\|\mathbf{a}_N\|_2 \le \sqrt{\sum_{n=0}^{\infty} a_n^2} = \|f\|_{L^2}.$$

This result answers a natural question: *why do we need the feature map to be an orthogonal basis of $L^2$?* In appendix E.1, we substantiate this question, also providing empirical evidence for the answer. Since $N < \infty$, we will incur a misspecification, which, for any $x \in \mathcal{X}$, corresponds to $\varepsilon(x) = f(x) - \langle \mathbf{a}_N, \boldsymbol{\varphi}_N(x) \rangle$. Nonetheless, we will show that this algorithm is able to perform very well in practice.

### 4.2 OB-PE

`OB-LinUCB` cannot achieve competitive regret since it does not explicitly handle the misspecification. For this reason, we propose a different choice for the linear bandit algorithm to be chosen as **LinBand**: the algorithm `phased elimination`, presented in Lattimore et al. (2020). With this modification, the algorithm can achieve a better regret guarantee, even if the misspecification is not negligible. We refer as `OB-PE` to the algorithm obtained by plugging `phased elimination` in Algorithm 1.

One feature of `phased elimination` Lattimore et al. (2020) is worth mentioning. This algorithm imposes that the number of arms is $k < \infty$, and its regret grows as $\log(k)$. Even if our set of arms $\boldsymbol{\varphi}_N \leftarrow [\varphi_0(x), \varphi_1(x) \dots \varphi_{\widetilde{N}}(x)] \; \forall x \in \mathcal{X}$ is uncountable, this does not represent an issue. Indeed, we can cover it by balls of radius $T^{-1/2}$ and preserve the same regret guarantee. This holds for two reasons: being $f$ Lipschitz continuous, making an $T^{-1/2}-$cover of $\mathcal{X}$ allows to retain an $LT^{-1/2}-$suboptimal arm. This translates in an additive term $+L\sqrt{T}$ on the regret, which is negligible with respect to the main part. The second reason is that $k$ grows as $\left(T^{1/2}\right)^d$, so that the multiplicative term on the regret corresponds to $(d/2)\log(T)$, which is also negligible.

## 5 MAIN RESULTS

In this section, we present the proofs of the main results regarding the regret bounds for algorithm 1 in different scenarios. We will always assume that the noise $\eta_t$ is i.i.d. and $\sigma-$subgaussian $\|f\|_\infty \le 1$ (one can be replaced by any constant by just rescaling). Both assumption are ubiquitous in the literature, and they strictly include the assumption to have an upper bound for $\|f\|_{L^2}$ done in algorithm 1, as by Hölder's inequality $\|f\|_{L^2} \le 2\|f\|_\infty \le 2$.

### 5.1 REGRET BOUND FOR $d = 1$

The first theorem provides a regret guarantee for algorithm `OB-PE`, which is optimal if $\mathcal{X} = [-1, 1]$. The same does not hold for higher dimension spaces since proof techniques are slightly different, as we shall see in the next subsection.

**Theorem 3.** *Fix $\delta > 0$ and assume that $f \in \mathcal{C}^s([-1, 1])$ and its $s + 1-$th derivative is square-integrable. With probability at least $1 - \delta$, algorithm OB-PE, when instantiated with Legendre or*

*Fourier[3] feature maps and $N = T^{\frac{1}{2s+1}}$ achieves regret*

$$R_T \leq \log(1/\delta)\,\widetilde{\mathcal{O}}(T^{\frac{s+1}{2s+1}}).$$

For the proof, see Appendix C.1. Taking $\delta = T^{-1}$, it follows that our algorithm has an expected regret that is bounded by $\widetilde{\mathcal{O}}(T^{\frac{s+1}{2s+1}})$. In is worth mentioning that the previous result hides in the $\widetilde{\mathcal{O}}(\cdot)$ notation constants depending on $f$, as in is for every algorithm for the continuous bandit setting. For Zooming, the regret depends on the Lipschitz constant of the function, for UCB-Meta-Algorithm on the Holder constant corresponding to the maximal degree of differentiability. For GP methods, the dependence is on the norm in the RKHS. This dependence on $f$ is unavoidable: since $\mathcal{C}^s(\mathcal{X})$ functions are dense in $\mathcal{C}^0(\mathcal{X})$, if it were possible to have a universal bound of order $T^{\frac{s+1}{2s+1}}$ valid for every $s-$times differentiable function without function-dependent constants, it would also be possible to extend the regret bound to the whole space $\mathcal{C}^0(\mathcal{X})$.

## 5.2 REGRET BOUND FOR $d > 1$

In the more general case of $\mathcal{X} = [-1, 1]^d$, we can prove another regret bound for algorithm OB-PE. Still, as we shall see, this bound is slightly suboptimal.

**Theorem 4.** *Fix $\delta > 0$ and assume that $f \in \mathcal{C}^s([-1, 1]^d)$. With probability at least $1 - \delta$, algorithm OB-PE, when instantiated with multivariate Legendre feature map (see 3.1) and $N = T^{\frac{d}{2s}}$ achieves regret*

$$R_T \leq \log(1/\delta)\,\widetilde{\mathcal{O}}(T^{\frac{2s+d}{4s}}).$$

This result has two relevant drawbacks. Firstly, it only holds for Legendre feature map, and not for Fourier ones. Second, its regret bound of order $T^{\frac{2s+d}{4s}}$ becomes vacuous when $d > 2s$. Still, this result is close to the known lower bound for the setting, telling that the regret cannot be smaller that $\Omega(T^{\frac{s+d}{2s+d}})$, so that in the regime $d > 2s$, even the optimal regret is very close to $\mathcal{O}(T)$. The reason why our regret is not optimal in this case are very deep, and we investigate them in the appendix E.2.

**Case of $f$ infinitely differentiable.** The previous results apply in the case of a smooth function of an arbitrary but finite degree $s$. It is also interesting to study the case of infinitely differentiable functions, for which we give a motivating example in Appendix D. In this case, we can achieve the best possible regret $\sqrt{T}$, up to logarithmic terms.

**Theorem 5.** *Fix $\delta > 0$ and assume that $f \in \mathcal{C}^\infty(\mathcal{X})$, being also analytic with convergence radius $1 + \rho$ for some $\rho > 0$. Then, with probability at least $1 - \delta$, algorithm OB-PE, when instantiated with multivariate Legendre feature map and $N = \log(T)^{\log(1+\rho)^{-1}}$, satisfies*

$$R_T \leq \log(1/\delta)\,\widetilde{\mathcal{O}}(\sqrt{T}).$$

This result imposes a stronger condition w.r.t. the fact of being just infinitely differentiable. In fact, it is required that the reward function $f$ is analytic in an open set containing $\mathcal{X}$. In fact, it could happen, for a $\mathcal{C}^\infty(\mathcal{X})$ function that is not analytic, that the approximation error relative to an $N$ degree feature map decreases more than polinomially but less than exponentially. Examples of functions of this kind are for example $N^{-\log(N)}$.

## 5.3 COMPARISON WITH STATE OF THE ART

In this section, we aim at comparing the result of this paper with the state of the art for continuous armed bandits. As anticipated in the introduction, there are many algorithms in the literature that are focused on the continuous-armed bandit problem, with many different ideas coming from different fields. As comparison, we have chosen the ones which achieve the best results for either regret or computational complexity, or are particularly popular. From the literature about Lipschitz bandits, we have chosen the celebrated Zooming (Kleinberg et al., 2008) and UCB-Meta-Algorithm (Liu et al., 2021), which achieves optimal regret bound. From the literature of Bayesian optimization, we

---

[3]in case we use Fourier basis, periodicity at the boundary of the interval

| Algorithm | $R_T(\mathcal{C}^s, \mathbb{R}^1)$ | $R_T(\mathcal{C}^s, \mathbb{R}^d)$ | $R_T(\mathcal{C}^\infty, \mathbb{R}^d)$ | Complexity |
|---|---|---|---|---|
| Zooming | $T^{2/3}$ | $T^{\frac{d+1}{d+2}}$ | $T^{\frac{d+1}{d+2}}$ | $k^2T$ |
| UCB-Meta-algorithm | $T^{\frac{s+1}{2s+1}}$ | $T^{\frac{s+d}{2s+d}}$ | N/A | $kT^2$ |
| IGP-UCB | $T^{\frac{s+3/2}{2s+1}}$ | $T^{\frac{s+3d/2}{2s+d}}$ | $T^{1/2}$ | $T^4 + kT^3$ |
| BPE | $T^{\frac{s+1}{2s+1}}$ | $T^{\frac{s+d}{2s+d}}$ | $T^{1/2}$ | $T^4 + kT^3$ |
| BBKB | $T^{\frac{s+3/2}{2s+1}}$ | $T^{\frac{s+3d/2}{2s+d}}$ | $T^{1/2}$ | $T^{\frac{4s+5d}{2s+d}} + kT^{\frac{4s+4d}{2s+d}}$ |
| mini-META | $T^{\frac{s+3/2}{2s+1}}$ | $T^{\frac{s+3d/2}{2s+d}}$ | $T^{1/2}$ | $T^2 + kT^{\frac{2s+4d}{2s+d}}$ |
| OB-PE (ours) | $T^{\frac{s+1}{2s+1}}$ | $T^{\frac{s+d/2}{2s}}$ | $T^{1/2}$ | $T^{\frac{3d}{2s}} + kT^{\frac{d}{s}}$ |
| Lower bounds | $\Omega(T^{\frac{s+1}{2s+1}})$ | $\Omega(T^{\frac{s+d}{2s+d}})$ | $\Omega(T^{1/2})$ | N/A |

Table 1: Algorithms and their theoretical performance. In the regret guarantees and in the computational complexity columns we have omitted the $\widetilde{\mathcal{O}}(\cdot)$ notation for readability.

have chosen IGP-UCB(Chowdhury & Gopalan, 2017), which gives the most optimized version of the celebrated GP-UCB algorithm and taken BPE (Li & Scarlett, 2022), which achieves optimal regret. Lastly, we have also considered two very recent algorithm BBKB and mini-META (Calandriello et al., 2020; 2022) from the field of Bayesian optimization which try to improve the computational efficiency of classical methods. We summarize the theoretical guarantees of these algorithms in Table 1. It is important to clarify some aspects that cannot be represented in the table.

**Assumptions on the regret guarantees**  Not all the assumptions on the regret are specified in the table. The algorithms from the literature of Bayesian Optimization (IGP-UCB, BPE, BBKB, mini-META), assume that the reward function $f$ belongs to an RKHS with kernel given by either the RBF (Gaussian) kernel or the Matérn one. To obtain the regret bound, we have used the well-known fact that the $\nu-$Matérn kernel contains functions that are $\lceil \nu \rceil$ times differentiable in the $L^2$ sense. However, this does *not* mean that the algorithm has a regret guarantee for any $\mathcal{C}^s$ function. Also our algorithm, for the case $R_T(\mathcal{C}^s, \mathbb{R}^1)$ requires to assume that the reward function has an $s + 1-$th derivative which is square-integrable.

**Computation of the time complexity**  In order to have a fair comparison between the algorithms, we have fixed some aspects that are shared. First, in the table we have reported, for each algorithm, the total time complexity to perform all the length $T$ episode. Since we are in continuous space, we need to assume that some form of discretization, otherwise it is impossible to choose the candidate arm at each round. We have assumed that all algorithms start from the same discretization of $\mathcal{X}$ which is composed of $k$ elements. Lastly, note that the complexities of BBKB, mini-META and OB-PE (ours) may seem to explode for $d \gg s$ while in fact this regime corresponds to an unfeasible region where the algorithms make linear regret.

Comparing our algorithm with the state of the art in the family of GP bandits, we can see that our OB-PE enjoys the same regret guarantees of the best of this family BPE, except for the case of a reward function on $\mathcal{X} = \mathbb{R}^d$ with $s < +\infty$, where our algorithm is slightly worse. Still, OB-PE wins with margin on computational complexity, where it is able to outperform even BBKB and mini-META.

UCB-Meta-algorithm (Liu et al. (2021)), unlike those based on Gaussian Processes, works under our same set of assumptions. As we have seen, this algorithm is able to outperform our regret guarantee in case of a reward function on $\mathcal{X} = \mathbb{R}^d$ with $s < +\infty$ but does not have a good regret guarantee in the $s = +\infty$ case, since their approach would require to make a linear bandit of infinite dimension. Furthermore, our algorithm outperfors UCB-Meta-algorithm on the computational side by a significant margin. Finally, OB-PE is arguably much simpler, as it requires to just build one global feature map instead of many local ones (a number which depends on $T$). This inherent simplicity enhances the algorithm's interpretability, allowing for the incorporation of domain knowledge. For instance, leveraging the fact that Legendre polynomials of even order are even functions, we can intelligently reduce the dimension of the feature map when we know that the objective function $f$ follows even or odd patterns, showcasing the flexibility of OB-PE.

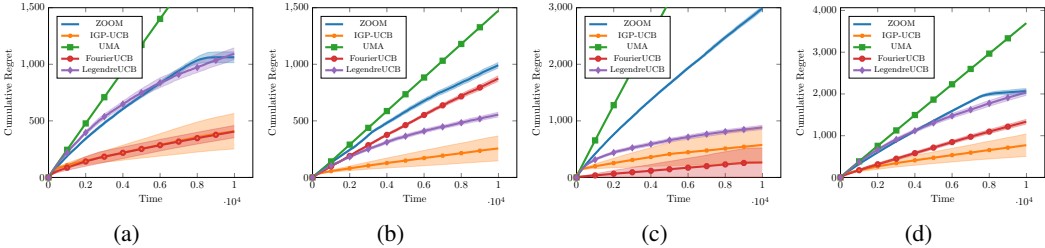

Figure 1: Regret plots of the algorithms in four environments (base version)

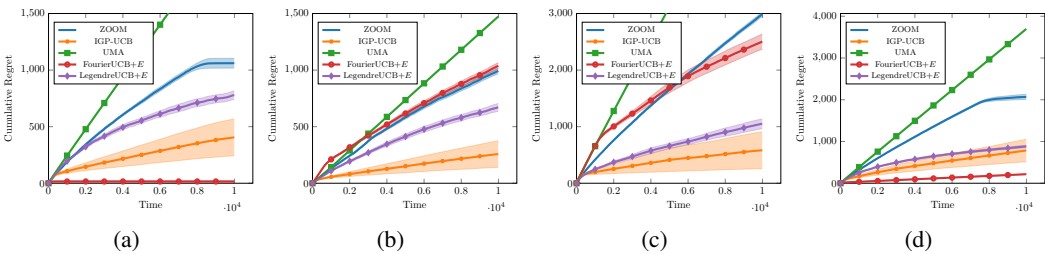

Figure 2: Regret plots of the algorithms in **the same** four environments (even version)

Overall, algorithm `OB-PE` is competitive with the state of the art under all aspects. We conclude this comparison with experiments over synthetic data to validate the considerations of this section.

## 6 EXPERIMENTS

To test the algorithms introduced in this paper, we have performed numerical simulations showing their performance compared to some baselines in the literature on continuous armed bandit environments. The details of the experiments are shown in Appendix F. Here we limit ourselves to a brief description of the results.

**Setting**  The various environments are characterized by the reward function $f$ since the noise added is always Gaussian with zero mean and unit variance. The choice of $f$, from left to right, corresponds to $(a)$ a Gaussian density function with $\sigma = 0.3$, $(b)$ An even polynomial of degree 4, $(c)$ A product between a $\sin$ function and a polynomial (not even neither odd), $(d)$ a triangular window function on the interval $[-1/2, 1/2]$. This function is piecewise linear and continuous, but not continuously differentiable. As baselines, both in Figures 1 and 2, we put the same algorithms of the literature appearing in the Table 1; `ZOOM` standing for `Zooming`, `IGP-UCB` for `IGP-UCB` and `UMA` for `UCB-Meta-algorithm`. As a comparison, in Figure 1, we have `OB-LinUCB` with the two sets of orthogonal functions, with the name `FourierUCB` and `LegendreUCB`. Instead, in Figure 2, we are showing the performance of the three algorithms when we only use *even* basis functions. This is done to test the flexibility of our algorithms in case they receive some information on the function $f$ we are optimizing, as it happens very often in practice due to the presence of domain knowledge. In experiment $(c)$, to test what happens when the algorithms receive a wrong advice, we consider a function $f$ that is not even.

**Results**  As we can see, the best-performing baseline is always `IGP-UCB`. This result comes with no surprise: this method is known to give extremely good results in practice. On the other side, `UMA` turns out to be the worst algorithm in every setting, despite having the best theoretical guarantees. To ensure the truthfulness of this result, we have conducted an extensive hyperparameter tuning of this baseline F.3, showing that even with the best parameters, the performance is not satisfactory. This can be explained by the fact that this algorithm comes from a very theoretical paper and was designed to prove a regret bound rather than to be valid in practice.

Table 2: Comparison of computation times for experiments on Environment $(a)$

| Algorithm | ZOOM | IGP-UCB | UMA | FourierUCB | LegendreUCB |
|-----------|------|---------|-----|------------|-------------|
| Time (s)  | 3.9  | 14957.3 | 108.1 | 3.2      | 3.4         |

Coming to our algorithms, we can see that LegendreUCB has stable performance, always performing better than ZOOM and always worse than IGP-UCB. The most surprising algorithm is indeed FourierUCB. This algorithm has weaker theoretical guarantees, as it requires periodic conditions at the boundary. Formally, this condition is satisfied only by Environment $(d)$ (which is $\mathcal{C}^0(\mathcal{X})$ and periodic) and partially by $(b)$ (which is $\mathcal{C}^\infty(\mathcal{X})$, but only $\mathcal{C}^0_{per}(\mathcal{X})$ since the derivative is not continuous at the boundary). Nonetheless, this algorithm is able to outperform the powerful IGP-UCB in half of the environments, both in the standard case and in the case with even basis functions. Its performance is particularly good in Environment $(a)$ of Figure 2. This can be explained by the fact that the Gaussian function with all its derivatives decreases to zero very quickly at the extremes of the interval. Therefore it can be well approximated by a $\mathcal{C}^\infty_{per}$ function with a value of 0 at the boundary.

Last but not least, it is important to consider the computational effort of the algorithms to perform the full experiment. In Table 2, we have reported the time to run all the experiments with Environment $(a)$. This conclusion agrees with our predictions in Table 1: the different time complexity leads to very different orders of magnitude in the actual running time, with IGP-UCB being five thousand times slower than our algorithms.

## 7 CONCLUSIONS

In this paper, we have introduced a new method to face the continuous armed bandit problem when the reward curve $f$ is $s$ times differentiable. To this end, we proposed to project the reward curve on the subspace generated by the first elements of an orthogonal basis of $L^2(\mathcal{X})$ (Fourier, Legendre). This representation allows to reduce the continuous armed bandit problem to a linear bandit with misspecification. This problem can be solved by means of the meta-algorithm 1, from which we have introduced the two algorithms: OB-LinUCB and OB-PE. As we have shown, the former is more oriented to practice, being simple and computationally efficient, while the latter has a regret guarantee that is close to the lower bound for reward functions $f \in \mathcal{C}^s(\mathcal{X})$, being able to match it if either $d = 1$ or $s = +\infty$. Moreover, this algorithm enjoys state-of-the-art computational complexity, even surpassing algorithms specifically designed to be fast. Finally, OB-LinUCB is validated in simulated environments where it achieves performance competitive with the best baseline with a significantly lower computational effort.

**Future works.**  In this paper we have presented an algorithm that bridges the gap between two families of methods, the one based on Gaussian processes and the one based on Lipschtzness/Hölder continuity. Despite having a strong similarity with UCB-Meta-algorithm, the proof techniques for the regret bounds are based on projecting an Hilbert space over a finite dimensional subspace, as the ones of Vakili et al. (2021) (the paper discovering the bound on $\gamma_T$ allowing for optimal regret in Gaussian process bandits). Therefore, the most interesting question is whether it is possible to reduce the two huge family of algorithms to a meta-algorithm which is able to achieve "best-of-both-worlds" performance.

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
