## A  INDEX OF THE NOTATIONS

In this section, we leave, for the reader's convenience, a table of the notations introduced in this paper.

| | |
|---|---|
| $\mathcal{X}$ | Space of arms |
| $d$ | Dimension of $\mathcal{X}$ in the sense of vector spaces |
| $T$ | Time horizon |
| $T$ | Cumulative regret |
| $f$ | The unknown reward function |
| $\mathcal{H}$ | Generic Hilbert space |
| $\langle \cdot, \cdot \rangle_{\mathcal{H}}$ | Scalar product in the Hilbert space $\mathcal{H}$ |
| $L^2(\mathcal{X})$ | Space of square-integrable functions over $\mathcal{X}$ |
| $\varepsilon(x)$ | Misspecification evaluated in $x$ |
| $\eta_t$ | Random noise |
| $\sigma$ | Subgaussianity constant of the noise |
| $\boldsymbol{\varphi}_N$ | generic feature map of degree $N$ |
| $\varphi_n$ | $n-$th element of a generic feature map $\boldsymbol{\varphi}_N$ |
| $a_n$ | Scalar product between $f$ and $\varphi_n$ |
| $\mathbf{a}_N$ | Vector of the coefficients relative to feature map $\boldsymbol{\varphi}_N$ |
| $\boldsymbol{\varphi}_{F,N}$ | Fourier feature map of degree $N$ (1 dimension) |
| $\boldsymbol{\varphi}_{L,N}$ | Legendre feature map of degree $N$ (1 dimension) |
| $\boldsymbol{\varphi}_{F,N}^d$ | Fourier feature map of degree $N$ ($d$ dimensions) |
| $\boldsymbol{\varphi}_{L,N}^d$ | Legendre feature map of degree $N$ ($d$ dimensions) |
| $N$ | Degree of a feature map |
| $\widetilde{N}$ | Lenght of a feature map ($= N$ if $d = 1$) |
| $\mathcal{C}^s(\mathcal{X})$ | $s-$times differentiable functions over $\mathcal{X}$ |
| $\mathcal{C}_{per}^s([-1,1])$ | $s-$times differentiable periodic functions on $[-1,1]$ |

## B  FORMAL STATEMENT OF THE DECAY PROPERTIES

The following theorems can be found in Quarteroni et al. (2010) Page 348 formula (9.30).

**Theorem 6.** *Note $\{\varphi_{L,n}\}_n$ as the Legendre feature map. Let $f : [-1,1] \to \mathbb{R}$ be a measurable function. If $f \in \mathcal{C}^s([-1,1])$, and $f^{(s+1)} \in L^2([-1,1])$,*

$$\left\| \sum_{n=0}^{N} a_{L,n} \varphi_{L,n} - f \right\|_{\infty} = C\|f\|_s N^{-s-1/2},$$

*where $C$ is a universal constant, while $\|f\|_s = \sum_{k=0}^{s+1} \|f^{(k)}\|_{L^2}$.*

Note that in the previous theorem, the continuity of the $s + 1$ derivative is not required, and in principle not even its boundedness. A similar result applies to the Fourier basis if we assume periodic conditions at the boundaries (Quarteroni et al. (2010) Page 365).

**Theorem 7.** *Let $f : [-1,1] \to \mathbb{R}$ be a measurable function. If $f \in \mathcal{C}_{per}^s([-1,1])$ (space of $s-$times differentiable functions which are periodic at the boundary), and $f^{(s+1)} \in L^2([-1,1])$,*

$$\left\| \sum_{n=0}^{N} a_{F,n} \varphi_{F,n} - f \right\|_{\infty} = C\|f\|_s N^{-s-1/2},$$

where $C$ is a universal constant, while $\|f\|_s = \sum_{k=0}^{s+1} \|f^{(k)}\|_{L^2}$

In the case of $f \in \mathcal{C}^\infty([-1,1])$, the bounds given in the previous theorems vanish more than polynomially in $n$, meaning that for any positive $k$ we have the following convergence property

$$\frac{\left\|f - \sum_{n=0}^N a_n \varphi_n\right\|_\infty}{N^{-k}} \to 0.$$

However, it should be noted that this result does not directly guarantee exponential decay in $N$, as we could encounter scenarios where the decay follows a function such as $N^{-\log(N)}$. To achieve exponential decay, it is necessary to assume analyticity of the function.

**Theorem 8.** *Wang & Xiang (2012) Let, $f : [-1,1] \to \mathbb{R}$ be a function that admits an analytic extension on the complex ellipse $\mathcal{E}_\rho$ given by*

$$\mathcal{E}_\rho := \left\{ z \in \mathbb{C} : z = \frac{u + u^{-1}}{2}, \ |u| = \rho > 1 \right\}.$$

*Then, the approximation with Legendre feature map satisfies*

$$\|f - \sum_{n=0}^N a_n \varphi_n\|_\infty \le 2\sqrt{\frac{\rho^2 + \rho^{-2}}{\rho^2 - 1}} \|f\|_\infty N^{1/2} \rho^{1/2 - N}.$$

A similar result holds for the approximation with Fourier feature map.

**Theorem 9.** *Katznelson (2004)(Exercise I.4.4) If $f$ is periodic and analytic, then there are $K(f), \beta(f) > 0$ such that*

$$a_{n,F} \le K(f) e^{-\beta(f)n}.$$

From this theorem, a straightforward corollary follows by summing all the terms $a_{F,n}$.

**Corollary 10.** *If $f$ is periodic and analytic, there are $K(f), \beta(f) > 0$ such that*

$$\left\|f - \sum_{n=0}^N a_{n,F}\varphi_{n,F}\right\|_\infty \le K(f)\frac{e^{-\beta(f)N}}{1 - e^{-\beta(f)}}.$$

This result guarantees that we can achieve exponentially accurate approximation using Fourier basis functions when the function is analytic and periodic.

### B.1 APPROXIMATION THEORY FOR MULTIVARIATE FUNCTIONS

Regarding the case of Legendre and Fourier feature maps, there is currently no result ensuring that the approximation of $f$ using the usual coefficients $a_{L,n} := \langle f, \varphi_{L,n} \rangle_{L^2}$ enjoys particular decay properties. Still, there are results showing that smooth functions can be well approximated by multivariate polynomials. The following result is a particular case of Theorem 1 by Bagby et al. (2002).

**Theorem 11.** *Let $f \in \mathcal{C}^s([-1,1]^d)$. Then, for every $N > 0$, there is a polynomial $p_N$ of degree at most $N$ such that*

$$\|f - p_N\|_\infty \le C(f)N^{-s},$$

*where $C(f)$ is a constant only depending on $f$.*

A stronger result holds for infinitely differentiable functions.

**Theorem 12.** *Let $f \in \mathcal{C}^\infty(\mathcal{X})$, be analytic with convergence radius $1 + \rho$ for some $\rho > 0$. Then, for every $N > 0$, there is a polynomial of degree at most $N$ such that*

$$\|f - p_N\|_\infty \le \frac{C(f)(1 + \rho)^{-N}}{\rho},$$

*where $C(f)$ is a constant only depending on $f$.*

*Proof.* Being $f$ analytic, we know that on the domain $\{x : \|x\|_\infty \le 1 + \rho\}$ we have

$$f(x) = \sum_{n=0}^{\infty} \sum_{|\alpha|=n} a_\alpha x^\alpha,$$

where $\alpha$ is one $d-$dimensional multi index with degree $n$. Since the convergence radius is $1 + \rho$, we know by root test (Boas, 2012)[4] that

$$(1+\rho)^{-1} = \limsup_n \left( \sum_{|\alpha|=n} |a_\alpha| \right)^{1/n}.$$

Using this result, there is a constant $C(f) < \infty$ such that $\sum_{|\alpha|=n} |a_\alpha| < C(f)(1+\rho)^{-n}$. Therefore, we have

$$\left\| f(x) - \sum_{n=0}^{N} \sum_{|\alpha|=n} a_\alpha x^\alpha \right\|_\infty = \left\| \sum_{n=N}^{\infty} \sum_{|\alpha|=n} a_\alpha x^\alpha \right\|_\infty$$

$$\le C(f) \sum_{n=N}^{\infty} (1+\rho)^{-n} \sup_{x \in [-1,1]^d} \|x\|_\infty^n$$

$$\le \frac{C(f)(1+\rho)^{-N}}{\rho}.$$

Therefore, the $N-$degree polynomial $\sum_{n=0}^{N} \sum_{|\alpha|=n} a_\alpha x^\alpha$ satisfies the thesis.

$\square$

## C  MISSING PROOFS

In this section, we provide the proof for the regret bounds of the main paper, relative to algorithm `OB-PE` under different assumptions on the reward function.

### C.1  REGRET BOUND FOR `OB-PE` IN THE $1-$DIMENSIONAL CASE

We now proceed to establish a regret bound for our second algorithm, `OB-PE`, which achieves a superior theoretical performance in terms of regret.

**Theorem 3.** *Fix $\delta > 0$ and assume that $f \in \mathcal{C}^s([-1,1])$ and its $s+1-$th derivative is square-integrable. With probability at least $1 - \delta$, algorithm `OB-PE`, when instantiated with Legendre or Fourier[5] feature maps and $N = T^{\frac{1}{2s+1}}$ achieves regret*

$$R_T \le \log\left(1/\delta\right) \widetilde{\mathcal{O}}(T^{\frac{s+1}{2s+1}}).$$

*Proof.* Due to the fact that in `OB-PE` is based on algorithm `phased elimination`, presented in Lattimore et al. (2020), we have to cover the space $[-1,1]$ to get a finite number of arms.

We can prove that starting form a discretization of the space $[-1,1]$ in $\lceil \sqrt{T} \rceil$ discrete arms has no effect on the regret.

$$R_T = \sum_{t=1}^{T} \max_{x \in [-1,1]} f(x) - f(x_t)$$

$$= \sum_{t=1}^{T} \max_{x \in [-1,1]} f(x) - \max_{j=1,\dots \lceil \sqrt{T} \rceil} f(x_j) + \max_{j=1,\dots \lceil \sqrt{T} \rceil} f(x_j) - f(x_t).$$

---

[4]see also this discussion (https://math.stackexchange.com/users/62443/johannes hahn)
[5]in case we use Fourier basis, periodicity at the boundary of the interval

Having assumed smoothness, we have also that $f$ is $L-$Lipschitz continuous for some $L > 0$. Therefore, the first part $\sup_{x \in [-1,1]} f(x) - \sup_{j=1,...\lceil \sqrt{T} \rceil} f(x_j)$ will be bounded by $LT^{-1/2}$. This results in

$$R_T \leq T \cdot LT^{-1/2} + \sum_{t=1}^{T} \max_{j=1,...\lceil \sqrt{T} \rceil} f(x_j) - f(x_t)$$

$$= \underbrace{LT^{1/2}}_{\in \widetilde{\mathcal{O}}(\sqrt{T})} + \sum_{t=1}^{T} \max_{j=1,...\lceil \sqrt{T} \rceil} f(x_j) - f(x_t).$$

Therefore, from now on, we will assume without loss of generality that the best arm belongs to our discretization $j = 1,...\lceil \sqrt{T} \rceil$. At this point, the algorithm requires to use `phased elimination` with a representation of the arms given, for every $j = 1,...\lceil \sqrt{T} \rceil$, by

$$\boldsymbol{\varphi}_N(x_j) \leftarrow [\varphi_0(x_j), \varphi_1(x_j) \ldots \varphi_N(x_j)].$$

Therefore, Proposition 5.1. by Lattimore et al. (2020) ensures the following high probability regret bound. For each $\delta > 0$, we have in the case of $k$ arms, with probability at least $1 - \delta$,

$$R_T \leq \log\left(\frac{1}{\delta}\right) \widetilde{\mathcal{O}}(\sqrt{NT} \log(k) + \varepsilon \sqrt{NT}), \tag{2}$$

where $\varepsilon$ is an upper bound on the misspecification, which in our case can be computed as follows:

$$\inf_{\boldsymbol{\theta} \in \mathbb{R}^N} \|f(x) - \langle \boldsymbol{\theta}, \boldsymbol{\varphi}_N(x) \rangle\|_\infty \leq \left\| f(x) - \sum_{n=0}^{N} a_n \varphi_n(x) \right\|_\infty$$

$$\leq C\|f\|_s N^{-s-1/2}$$

$$= C\|f\|_s T^{\frac{-s-1/2}{2s+1}} = C\|f\|_s T^{-\frac{1}{2}}.$$

where the second passage comes from theorem 6 (theorem 7 in case of Fourier basis, and $C$ is the universal constant there defined) and the third from the definition $N = T^{\frac{1}{2s+1}}$. Substituting this value to the maximal misspecification $\varepsilon$, in equation equation 2 the regret is bounded with probability $1 - \delta$ by

$$R_T \leq \log\left(\frac{1}{\delta}\right) \widetilde{\mathcal{O}}(\sqrt{NT} \log(k) + \sqrt{N}\sqrt{T}) = \log\left(\frac{1}{\delta}\right) \widetilde{\mathcal{O}}(T^{\frac{s+1}{2s+1}}),$$

where the last passage is valid since, due to our discretization the number of arms corresponds to $\lceil \sqrt{T} \rceil$, so that $\log(k) \approx \log(T)/2$. This step ends the proof. $\qquad \square$

## C.2 REGRET BOUND FOR $d > 1$

In this section, we prove the main result for our algorithm in the multivariate case $\mathcal{X} = [-1, 1]^d$.

**Theorem 4.** *Fix $\delta > 0$ and assume that $f \in \mathcal{C}^s([-1,1]^d)$. With probability at least $1 - \delta$, algorithm* `OB-PE`*, when instantiated with multivariate Legendre feature map (see 3.1) and $N = T^{\frac{d}{2s}}$ achieves regret*

$$R_T \leq \log(1/\delta) \widetilde{\mathcal{O}}(T^{\frac{2s+d}{4s}}).$$

*Proof.* We start from the proof of the result for one dimension, changing only what is needed.

1. In order have apply the algorithm `phased elimination` and the desired regret bound, we need to make a $T^{-1/2}-$cover of the state space, so that the number of actions becomes

finite. In the one dimensional case, this cover contained $\lceil\sqrt{T}\rceil$ points, while here we need $k = \lceil\sqrt{T}\rceil^d$. Still, since this number enters in the regret just as $\log(k)$, this only makes it scale by a factor $d$.

2. In the multivariate case, the dimension $\widetilde{N}$ of the feature vector does not coincide with the degree of the polynomial obtained. In fact, to have a vectors which forming a basis for the vector space of degree $d-$variate polynomials of degree $N$, we need $\widetilde{N} = \binom{N+d}{N}$. This quantity is always bounded by $N^d$, which is much easier to compute.

3. In the multivariate case, we have no result about the decay property of the coefficients of Legendre polynomials. Therefore, we cannot bound $\|f(x) - \sum_{n=0}^{N} a_n \varphi_n(x)\|_\infty$ as done for the univariate case. Still, using `phased elimination` we are not forced to choose $\boldsymbol{\theta} = \mathbf{a}_N$, the vector of Legendre coefficients. Instead, we can rely on the following argument: due to theorem 11, we know that for every $N > 0$, there is a polynomial $p_N$ of degree at most $N$ such that

$$\|f - p_N\|_\infty \leq C(d)N^{-s}, \tag{3}$$

where $C(d)$ is a constant only depending on $d$. Therefore, since the feature map $\boldsymbol{\varphi}_{F,N}^d$ forms a basis for the vector space of all $d-$variate polynomials of degree at most $N$, this means that there is a vector $\boldsymbol{\theta}_*$ such that $\langle\boldsymbol{\varphi}_{L,N}^d(x), \boldsymbol{\theta}_*\rangle = p_N(x)$, the polynomial defined in equation equation 3. Then, we have

$$\inf_{\boldsymbol{\theta}\in\mathbb{R}^{\widetilde{N}}} \|f(x) - \langle\boldsymbol{\theta}, \boldsymbol{\varphi}_{L,N}^d(x)\rangle\|_\infty \leq \|f(x) - \langle\boldsymbol{\theta}_*, \boldsymbol{\varphi}_{L,N}^d(x)\rangle\|_\infty$$
$$= \|f - p_N\|_\infty$$
$$\leq C(d)N^{-s}.$$

Once done these three modifications, the proof follows similarly: once chosen $N = T^{\frac{1}{2s}}$ we get $\widetilde{N} \leq N^d = T^{\frac{d}{2s}}$. Having proved that the maximal misspecification $\varepsilon$ is bounded by $C(f)N^{-s}$, the regret is bounded with probability $1 - \delta$ by

$$R_T \leq \log\left(\frac{1}{\delta}\right) \widetilde{\mathcal{O}}\left(\sqrt{\widetilde{N}T}\log(k) + \sqrt{\widetilde{N}}TC(f)N^{-s}\right)$$
$$\leq \log\left(\frac{1}{\delta}\right) \widetilde{\mathcal{O}}\left(\sqrt{(T^{\frac{d}{2s}})T}\log(k) + (T^{\frac{d}{4s}})TC(f)T^{-\frac{1}{2}}\right)$$
$$= \log\left(\frac{1}{\delta}\right) \widetilde{\mathcal{O}}(T^{\frac{2s+d}{4s}}).$$

where the last passage is valid since, due to our discretization the number of arms corresponds to $\lceil\sqrt{T}\rceil^d$, so that $\log(k) \approx d\log(T)/2$. This step ends the proof.

$\square$

### C.3 Regret bounds in the case of $\mathcal{C}^\infty$ functions

**Theorem 5.** *Fix $\delta > 0$ and assume that $f \in \mathcal{C}^\infty(\mathcal{X})$, being also analytic with convergence radius $1 + \rho$ for some $\rho > 0$. Then, with probability at least $1 - \delta$, algorithm `OB-PE`, when instantiated with multivariate Legendre feature map and $N = \log(T)^{\log(1+\rho)^{-1}}$, satisfies*

$$R_T \leq \log\left(1/\delta\right) \widetilde{\mathcal{O}}(\sqrt{T}).$$

*Proof.* It is sufficient to repeat the same passages of the previous proof with the bound of the misspecification given by theorem 12. $\square$

# D    APPLICATION: TREMBLING HAND PROBLEM

The smoothness assumption upon which our algorithm is built may appear overly restrictive. In practice, it becomes challenging to ensure that the function $f$ in the continuous armed bandit problem exhibits a certain degree of smoothness. Hence, we propose a modified setting where the condition naturally holds.

Up until now, our assumption has been that by selecting an arm $x_t \in [-1, 1]$, we could observe a sample $f(x_t) + \eta_t$, where $\eta_t$ represents a zero-mean noise independent of the past. However, a more realistic scenario involves an additional unobservable noise $\zeta_t$ (also i.i.d.) affecting the choice of the arm $x_t$. In fact, in a continuous state space, it is likely that the choice of actions is affected by some sort of random error due, for example, to measurement errors. A similar model is known in the field of game theory as the "trembling hand problem" Jackson et al. (2012). Here, we make two key assumptions. First, we assume that the 'horizontal noise' $\zeta_t$ follows a Gaussian distribution. Second, we assume that the function $f \in L^\infty_{per}([-1, 1])$, allowing for the evaluation of the function outside the interval $[-1, 1]$ through periodicity. This assumption is necessary due to the Gaussian nature of the noise $\zeta_t$, which requires the function $f$ to be defined on the entire space $\mathbb{R}$. Importantly, no specific smoothness assumptions are imposed on $f$.

In this setting, it is meaningless to compare to the fixed and noiseless best arm, since it would lead to linear regret because of the noise. We can instead compare the policy to the best arm when the noise is applied. This leads to the following definition of regret

$$R_T := \sum_{t=1}^{T} \sup_{x \in [-1,1]} \mathbb{E}_{\zeta \sim \mathcal{N}(0,\sigma^2)} f(x + \zeta) - f(x_t + \zeta_t) \qquad \zeta_t \overset{i.i.d.}{\sim} \mathcal{N}(0, \sigma^2).$$

This kind of problem can be reduced to a standard continuous bandit problem by just substituting the reward curve $f$ with

$$\tilde{f} := f * \mathcal{N}(0, \sigma^2) = \int_0^\infty f(x - y) \frac{e^{-y^2/(2\sigma^2)}}{\sqrt{2\pi}\sigma} \, dy.$$

In this way, the objective function $f$ gets smoothed by the effect of the noise $\zeta$; it is well known that the convolution between a $\mathcal{C}^\infty$ function, such as the Gaussian density function, and a bounded function, such as $f$, is $\mathcal{C}^\infty$ too. This allows us to prove that the Fourier coefficients $a_n$ of $\tilde{f}$ decay exponentially fast. Moreover, in this particular case, it is possible to provide a more precise quantification of their decay.

**Theorem 13.** *The Fourier coefficients of $\tilde{f}$ are bounded as follows*

$$|a_n| \leq \|f\|_{L^2} \times \begin{cases} e^{-\sigma^2 n^2/8} & n \text{ even} \\ e^{-\sigma^2 (n+1)^2/8} & n \text{ odd} \end{cases}$$

*Proof.* First of all, note that using the Fourier series in exponential form, we have

$$f(x) = \sum_{n=-\infty}^{\infty} b_n e^{inx},$$

for a sequence $b_n$, which respects Parseval's identity:

$$\sum_{n=-\infty}^{\infty} b_n^2 = \|f\|_{L^2}^2 < +\infty. \tag{4}$$

Now, note that

$$\tilde{f}(x) = \int_{-\infty}^{\infty} f(x-y) \frac{e^{-y^2/(2\sigma^2)}}{\sqrt{2\pi}\sigma} \, dy$$

$$= \int_{-\infty}^{\infty} \sum_{n=-\infty}^{\infty} b_n e^{in(x-y)} \frac{e^{-y^2/(2\sigma^2)}}{\sqrt{2\pi}\sigma} \, dy$$

$$= \sum_{n=-\infty}^{\infty} e^{inx} b_n \int_{-\infty}^{\infty} e^{-iny} \frac{e^{-y^2/(2\sigma^2)}}{\sqrt{2\pi}\sigma} \, dy.$$

Observe that the last term corresponds to the Fourier transform of $\frac{e^{-y^2/(2\sigma^2)}}{\sqrt{2\pi}}$ evaluated at $n$. Since the Fourier transform of $\frac{e^{-y^2/(2\sigma^2)}}{\sqrt{2\pi}}$ corresponds to $e^{-\sigma^2\xi^2/2}$, this term corresponds to $e^{-\sigma^2 n^2/2}$. Substituting, we get

$$\tilde{f}(x) = \sum_{n=-\infty}^{\infty} b_n \int_{-\infty}^{\infty} e^{iny} \frac{e^{-\sigma^2 y^2/(2\sigma^2)}}{\sqrt{2\pi}\sigma} \, dy$$

$$= \sum_{n=-\infty}^{\infty} e^{-\sigma^2 n^2/2} b_n e^{inx}.$$

This means that $b'_n = e^{-\sigma^2 n^2/2} b_n$ corresponds to the Fourier coefficients of $\tilde{f}$ in exponential form. By equation equation 4

$$b'_n \leq e^{-\sigma^2 n^2/2} \|f\|_{L^2}.$$

To obtain a bound for the Fourier coefficients in sin-cosine form, it is sufficient to apply the trigonometric identities

$$\cos(x) = \frac{e^{ix} + e^{-ix}}{2} \qquad \sin(x) = \frac{e^{ix} - e^{-ix}}{2i}.$$

$\square$

This fact leads to the conclusion that both algorithms `OB-LinUCB` and `OB-MissLinUCB` are able to achieve regret $\tilde{\mathcal{O}}(\sqrt{T})$ regret in this case. To see this, it is sufficient to substitute the decay rate $e^{-\sigma^2 n^2/8} \|f\|_{L^2}$, which is faster than exponential, in the proof of theorem 5.

## E    FURTHER CONSIDERATIONS

In this section, we address some question of theoretical relevance, which we cannot insert in the main paper due to the limited space.

### E.1    DOES `OB-LINUCB` WORK WITH NONORTHOGONAL FEATURES?

As linear bandit algorithms are designed to work whenever the reward can be written as a scalar product $r_t = \langle \boldsymbol{\theta}, \boldsymbol{\varphi}(x) \rangle + \eta_t$, one could do the following reasoning. Since the vector space generated by the first $N$ Legendre polynomials $\varphi_{L,1}(x), \dots \varphi_{L,N}(x)$ corresponds to the one generated by the basis $1, x, \dots x^N$, we can use the former as feature vector and still achieve the same results.

This reasoning may seem correct, but in fact it is affected by a subtle problem. The `LinUCB` algorithm, in order to work properly, needs not only that the average rewards have the form $\langle \boldsymbol{\theta}, \boldsymbol{\varphi}(x) \rangle$, but also that an upper bound on $\|\boldsymbol{\theta}\|_2$ is known. While this assumption is natural in the linear bandit setting, in the continuous armed bandit one, our assumption only includes a bound for $\|f\|_\infty$, which is not necessary linked to a bound on $\|\boldsymbol{\theta}\|_2$. While in the main paper it is shown that, for an orthogonal feature map $\|\boldsymbol{\theta}\|_2 \leq 2\|f\|_\infty$, the same result does not hold in general.

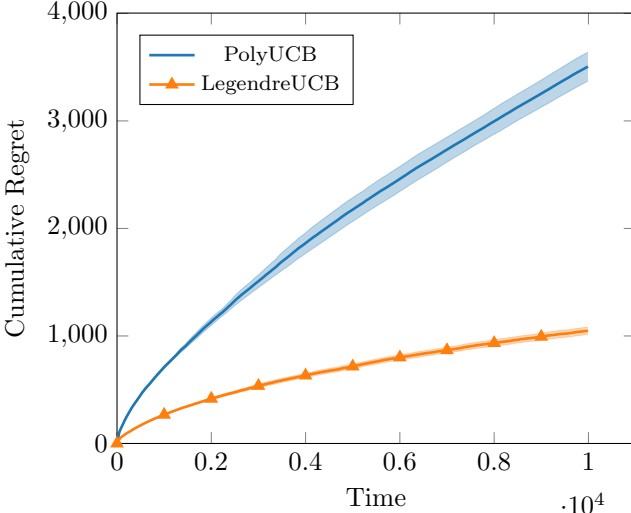

Figure 3: Regret curves of `LegendreUCB` and its trivial version `PolyUCB` which uses the standard polynomial basis instead of the Legendre basis.

Coming back to our specific case, where the feature map $\{1, x, \ldots x^N\}$ is proposed, we can see clearly this phenomenon. In fact, there are functions such that $\|f\|_\infty = \sup_{x \in [-1,1]} |f(x)| \leq 1$ but its corresponding $\theta$ coefficient has huge norm. For example, consider the polynomial:

$$p(x) = -0.3652 + 12.3076x^2 - 64.974x^4 + 109.5056x^6 - 57.4752x^8.$$

Even having very big coefficients in the base formed by $\{1, x, \ldots x^N\}$, it satisfies $\|p\|_\infty = 1$. Using this function as reward function, we can make an experiment to compare our `LegendreUCB` with the algorithm obtained by using the basis $\{1, x, \ldots x^N\}$ in `OB-LinUCB`, which we call `PolyUCB`. In figure 3, we have shown the result of this experiment. As we can see, `OB-LinUCB` is able to completely outperform its competitor, and the reason stays in the fact that polynomial $p(x)$ can be represented as a linear combination of the elements of the Legendre basis with $\theta$ is bounded by 2, while this does not hold for the standerd polynomial basis.

### E.2 Connections with recent advances in Gaussian Process Bandits and tightness of the regret

Despite our approach resembling that of Liu et al. (2021), one of the most intriguing aspects of our analysis lies in its parallelism with one of the most significant papers in the Gaussian process bandit field, namely Vakili et al. (2021). At a high level, the idea behind our proofs is to project the subspace of $s$-times differentiable functions onto the vector space formed by fixed-degree algebraic or trigonometric polynomials. Similarly, Vakili et al. (2021) accomplishes, albeit in different terms, a projection of an RKHS onto the vector space generated by the first $N$ eigenfunctions of the kernel.

At this point, both our work and theirs rely on analogous results: while we utilize the results from the theory of approximation B to bound the projection error, they employ the rate of decay of kernel eigenvalues. It is worth noting that, even though the space of $s$-times differentiable functions does not admit a kernel, we can view sequences of orthogonal functions as analogs of kernel eigenfunctions and the results on the decay of kernel eigenvalues as analogs of the decay properties of Fourier/Legendre coefficients B.

Now the question is as follows: if the results are analogous, how is it possible that in the case of Gaussian process bandits, the article by Vakili et al. (2021) enables achieving the optimal regret of the multidimensional case, namely $T^{\frac{s+d}{2s+d}}$, while our regret bound is slightly worse?

To understand the reason, we need to analyze the regret of Gaussian process bandits:

$$R_T \leq \widetilde{\mathcal{O}}(\sqrt{T} \underbrace{\sqrt{\widetilde{N} + \delta_{\widetilde{N}} T}}_{\sqrt{\gamma_T}})$$

This follows from equation (7) by Vakili et al. (2021) (here written in our notation) and the fact that the optimal regret for GP bandit is $\widetilde{\mathcal{O}}(\sqrt{T\gamma_T})$. The final bound follows from the fact that $\delta_{\widetilde{N}}$ can be proved to be of order $\widetilde{N}^{1-\frac{2s+d}{d}}$, so that the optimal regret corresponds to, optimizing the value of $\widetilde{N} = T^a$,

$$R_T \leq \widetilde{\mathcal{O}} \left( T^{\frac{1}{2}+\frac{1}{2}\min_a \max\{a, 1-\frac{2sa}{d}\}} \right).$$

For $a = \frac{d}{2s+d}$, this achieves the optimal bound $T^{\frac{s+d}{2s+d}}$. Instead, from the proof of our theorem 4 we have that the order of our regret can be written as

$$R_T \leq \widetilde{\mathcal{O}} \left( \sqrt{\widetilde{N}T} + \sqrt{\widetilde{N}}TN^{-s} \right)$$
$$= \widetilde{\mathcal{O}} \left( T^{\frac{1}{2}+\frac{1}{2}\min_a \max\{a, 1+\mathbf{a}-\frac{2sa}{d}\}} \right).$$

Where the only difference is the additional term $a$ in the last exponent, which is highlighted in red. Not coincidentally, this term arises precisely at the point where our approach and that of Vakili et al. (2021) diverge. While in their case, Gaussian process properties are applied, in ours, we are compelled to use a misspecified bandit algorithm to account for projection error. Presently, the most prominent algorithm known for this problem is the "phased elimination" algorithm, introduced by Lattimore et al. (2020). This algorithm provides a bound on the regret for bandits with a maximum misspecification $\varepsilon$ and dimension $D$ in the form:

$$R_T = \widetilde{\mathcal{O}} \left( \sqrt{DT} + \sqrt{\mathbf{D}}T\varepsilon \right).$$

Where the term in red is precisely what prevents our regret to be optimal. The existence of this term has been recognised as very annoying by the same authors of Lattimore et al. (2020), even if it cannot be eliminated for some feature maps. In the end, there are essentially three possibilities: 1) the optimal regret for Gaussian process bandits was proved to be $T^{\frac{s+d}{2s+d}}$ only thanks to the strong assumption of being in an RKHS and methods based on projection cannot achieve the optimal regret for general $\mathcal{C}^s(\mathcal{X})$ spaces 2) the regret bound of our algortihm can be refined to be $T^{\frac{s+d}{2s+d}}$ by improving the regret bound of Lattimore & Szepesvári (2020) in case of our specific feature maps 3) It is possible to modify our approach in a way that does not involve misspecified bandits to achieve regret $T^{\frac{s+d}{2s+d}}$. We leave this question as an open problem.

## F    EXPERIMENTS ADDENDA

In this section, we provide a detailed explanation of the experiments conducted in the main paper. Due to space limitations, we were unable to fully elaborate on the rationale behind the selection of the environments discussed in the main paper. Therefore, this section primarily aims to address this gap and provide a comprehensive understanding of the chosen environments.

### F.1    ENVIRONMENTS

The four plots depicting the function $f$ for each of the four environments can be observed in plots 4 and 5. It is worth noting that the applied noise is consistently Gaussian with a variance of 1.0. Some consideration on the four choices follow.

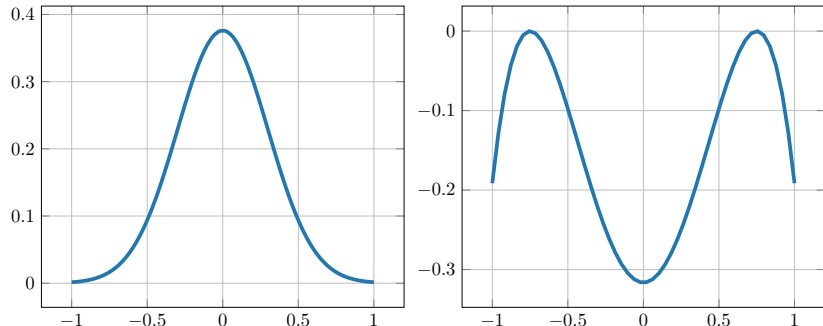

Figure 4: Left: experiment $(a)$, Right: experiment $(b)$.

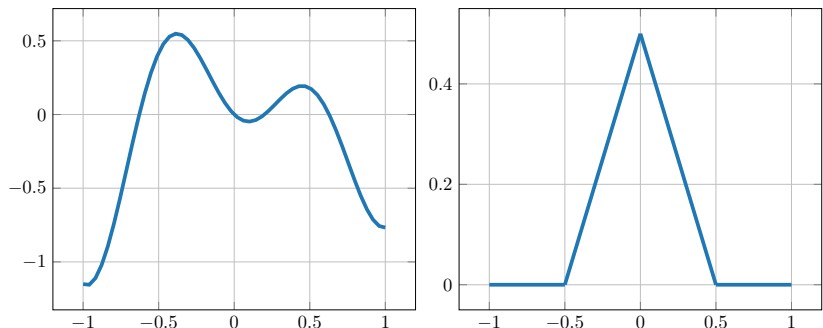

Figure 5: Left: experiment $(c)$, Right: experiment $(d)$.

(a) This curve is a Gaussian, belonging to $\mathcal{C}^{\infty}([-1, 1])$, so it is ideal for algorithms `LegendreUCB` and `ChebyshevUCB`. Instead, if we look at the periodicity at the boundary, this function is only in $\mathcal{C}^0_{per}([-1, 1])$, which seems not to be the best scenario for `FourierUCB`. Nonetheless, as the derivatives of any order are small at the boundaries of the interval, we can conjecture that the function is can be well approximated by a $\mathcal{C}^{\infty}_{per}([-1, 1])$. Moreover, this function is even, thus the version of the algorithms with domain knowledge (called $+E$ in the experiments) should perform better.

(b) This curve is a polynomial of degree four, as before $f \in \mathcal{C}^{\infty}([-1, 1])$. As before, if we look at the periodicity at the boundary, this function is only in $\mathcal{C}^0_{per}([-1, 1])$, but differently from the Gaussian case, we have $f'(-1) \gg f(1)$ which makes it very hard for `FourierUCB` to work. As the previous one, this function is even.

(c) This curve is a product between a sinusoidal wave and a polynomial, so again $f \in \mathcal{C}^{\infty}([-1, 1])$. This time the function is not in $\mathcal{C}^0_{per}([-1, 1])$, which means `FourierUCB` has no performance guarantee. This function, differently from the other ones, is not even, so the algorithms with domain knowledge like `ChebyshevUCB`$+E$ are receiving a wrong information, and are not guaranteed to work.

(d) This curve is piecewise linear, $1-$Lipschtz but only in $\mathcal{C}^0([-1, 1])$. However, it is also in $\mathcal{C}^0_{per}([-1, 1])$ which means `FourierUCB` has the same performance guarantee of `LegendreUCB` in this case. This function is also even.

As demonstrated in the main paper, the experimental results largely align with the predictions of the theoretical analysis, with one notable exception. Environment $(c)$ stands out due to its discontinuity at the boundaries, which renders `FourierUCB` without a performance guarantee in this particular setting. However, contrary to expectations, plot $(c)$ of Figure 1 illustrates that `FourierUCB` achieves remarkable performance, even surpassing the performance of `IGP-UCB`.

### F.2 HYPERPARAMETERS OF THE ALGORITHMS

In this section, we provide a comprehensive overview of the hyperparameters utilized for each of the algorithms employed in experiments 1 and 2.

1. `ZOOM`. This algorithm, referred to as Zooming, does not have any specific hyperparameters. Its implementation follows that of Kleinberg et al. (2008), with the exception of the covering oracle, which can be simplified since we are working in one dimension.

2. `IGP-UCB`. For IGP-UCB, we have followed the instructions of Chowdhury & Gopalan (2017). First, $R = 1$, due to the fact that the noise is Gaussian, and in particular $1-$subgaussian. As we are in dimension 1, we have put $\gamma_t = \log(t)^2$, while for the norm in the RKHS, we have $B = 4$. The confidence is set to $\delta = 1/T$, even if choosing $\delta = 1/\sqrt{T}$ gives similar results. Lastly, due to problems of stability we have imposed to evaluate every point 10 times. This procedure is also well known to mitigate computational issues significantly reducing the running time. For the kernel we have used the standard Radial Basis Functions, since most of the benchmarks are infinitely differentiable.

3. `UMA`. For the UCB-Meta-algorithm Liu et al. (2021), we encountered poor performance across all environments. As a result, we dedicated a separate subsection F.3 to thoroughly tune the algorithm and investigate the reasons behind its underperformance.

4. For our algorithms, `FourierUCB`, `LegendreUCB` and `ChebyshevUCB` we have two Hyperparameters to consider. The first hyperparameter is $m$, which serves as an upper bound on the $L^2$ norm of $f$. For `FourierUCB`, we set $m = 0.1$, while for `LegendreUCB` and `ChebyshevUCB`, we set $m = 1.0$. These values are reasonable, taking into account the magnitudes of the benchmark functions. The second hyperparameter is $N$, which determines the number of features to include. For `FourierUCB`, we set $N = 8$, and for `LegendreUCB` and `ChebyshevUCB`, we set $N = 6$.

### F.3 HYPERPARAMETER TUNING OF `UMA`

To prove the truthfulness of the results of the experiment in the main paper, which show a terrible performance for `UCB-Meta-algorithm`, we have tuned its hyperparameters in a separate experiment on our benchmark $(a)$, for a reduced time horizon $T = 5000$. The parameters used to perform the experiment in the main paper are already the best found.

The `UCB-Meta-algorithm` (Liu et al., 2021) allows for tuning three different hyperparameters:

`alpha`: This parameter represents the degree of the Taylor series utilized by the algorithm. `bins`: It determines the number of bins into which the interval $[-1, 1]$ is divided. `epsilon`: This hyperparameter serves as an upper bound on the value of misspecification.

After fixing a specific set of values for each hyperparameter, namely `alpha` $\in 4, 5, 6, 8, 10, 12$, `bins` $\in 5, 8, 10, 20$, and `epsilon` $\in 0.0, 0.01, 0.05, 0.1$, we conducted a random search by sampling 50 tuples of hyperparameters from the defined sets. For each tuple, we evaluated the algorithm with 5 different random seeds on environment $(a)$. Finally, we selected the tuple that yielded the best performance among the evaluated tuples.

**Results**    The best performing tuple of hyperparameters reveals to be `alpha` $= 4$, `bins` $= 8$ and `epsilon` $= 0.1$. Still, its performance is not significantly different from the others. In the following plot 6, we show the regret obtained by the algorithms as a functions of the three hyperparameters, for the points considered.

We can see that the performance seems to improve for small values of `alpha` and `bins` while it does not change significantly with `epsilon` ($z-$axis). However, the range between the best values in bright red and the worst in blue is narrow, just 1179.7 versus 1317.0.

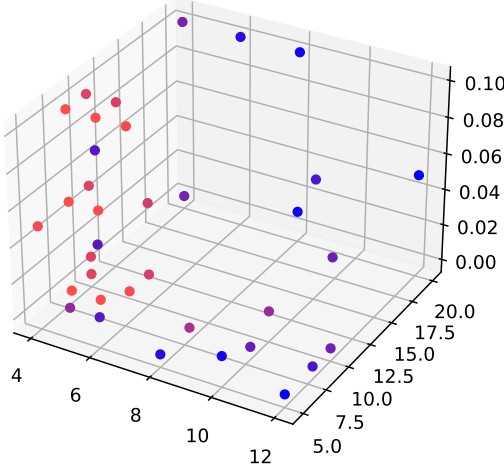

Figure 6: Regret of the algorithm depending on the hyperparamters chosen. On the three axes we have `alpha` (left axis), `bins` (right axis) and `epsilon` ($z-$axis). Bright red = better, Deep blue = worse.

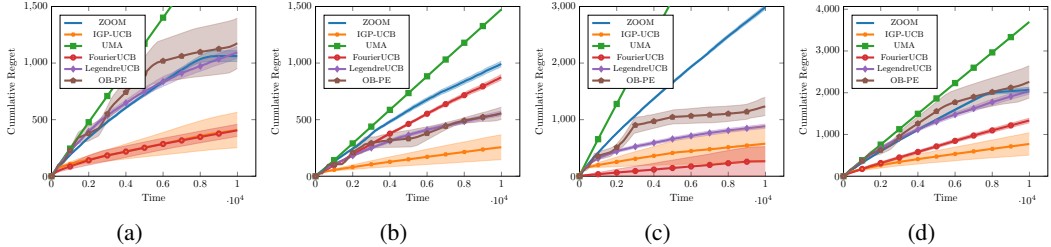

Figure 7: Regret plots of the algorithms in four environments (base version)

### F.4 ADDITIONAL EXPERIMENTS

The new experiments, whose regret curves are plotted in 7, are done on the same four environments, but adding the `OB-PE` baseline. This algorithm has only been run with Legendre basis function, as the trigonometric basis is endowed of slightly less theoretical guarantees. As we can see, on one side `OB-PE` surpasses the algorithm with the best theoretical guarantees, `UMA` and, in some environments, `Zooming`. On the other hand, we can see how this algorithm is outperformed by our `LegendreUCB`, which has no theoretical guarantees. As very often happens, in practice the simplicity of `LegendreUCB` seems to win over the most involved, but theoretically grounded, `OB-PE`. Running times of the algorithms can be found in table 3.

**Experiment in 2D** To be complete, as the experiments in the main paper only deal with continuous bandits on $[-1, 1]$, we performed an experiment with the action space being $[-1, 1]^2$. In this case, the reward function corresponds to $f(\boldsymbol{x}) = e^{-\|\boldsymbol{x}\|_2^2} = e^{x_1^2 + x_2^2}$. Results in figure 8 show that, as before, `LegendreUCB` is able to achieve a much better regret than `OB-PE`. Still, the latter shows a regret curve that significantly flattens after roughly $8000$ time-steps. This suggests that the former algorithm could be superior for very long horizons. Running times of the algorithms can be found in table 4.

Table 3: Comparison of computation times for experiments of figure 7 on Environment $(a)$

| Algorithm | ZOOM | IGP-UCB | UMA | FourierUCB | LegendreUCB | OB-PE |
|-----------|------|---------|-----|------------|-------------|-------|
| Time (s)  | 3.9  | 14957.3 | 108.1 | 3.2 | 3.4 | 1.1 |

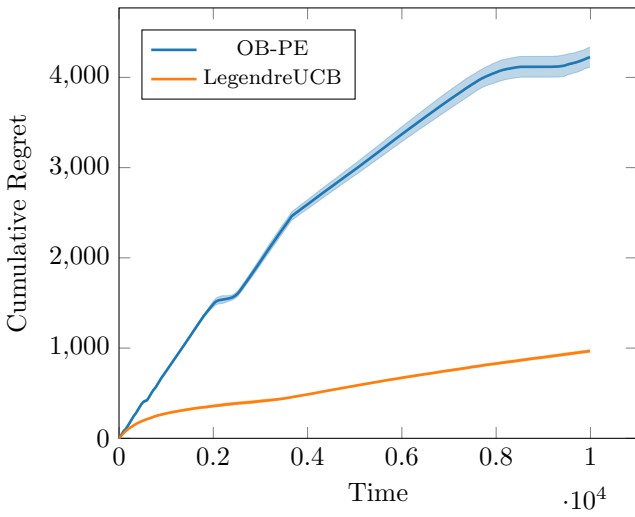

Figure 8: Regret curves of `LegendreUCB` and `OB-PE` on an environment with two dimensions.

### F.5 DETAILED EXPLANATION OF THE EXPERIMENTS

In this section, we report all the details of the experiments performed in the paper. These are important to ensure the truthfullness of the results and the claims based on empirical validation.

**Training Details**  In the main paper, we presented a total of eight experiments, with two experiments conducted for each distinct environment. Each experiment was executed using twenty random seeds, and the computations were distributed across twenty parallel processes using the `joblib` library. The total computational time for each experiment closely aligns with the running time of the slowest algorithm, which in this case is the `IGP-UCB` algorithm. As stated in the main paper, the running time for the IGP-UCB algorithm was measured to be 14957.3 seconds, approximately four hours.

**Compute**  We used a server with the following specifications:

- **CPU**: `88 Intel(R) Xeon(R) CPU E7-8880 v4 @ 2.20GHz cpus`
- **RAM**: `94,0 GB`

As mentioned, we parallelized the computing for the 20 different random seeds, therefore only 20 of the 88 cores were actually used.

**Reproducibility**  Given the stochastic nature of the bandit problem, we conducted multiple simulations to account for variability. All experiments were repeated with a total of 20 different random seeds, corresponding to the first 20 natural numbers. The random seed influenced the generation of rewards by the environment, while the proposed algorithms, being deterministic, were unaffected

Table 4: Running times for the experiment in figure 8

| Algorithm | LegendreUCB | OB-PE |
|-----------|-------------|-------|
| Time (s)  | 15.2        | 85.8  |

by the seed. This approach allowed us to capture the performance of the algorithms across different random realizations of the bandit problem.