# OpenReview forum: "Orthogonal Function Representations for Continuous Armed Bandits"
_ICLR.cc/2024/Conference — Submitted to ICLR 2024_

### Official Review · Reviewer_DACT · 2023-10-13

**Soundness:** 2 fair
**Presentation:** 2 fair
**Contribution:** 2 fair
**Rating:** 5
**Confidence:** 3

**Summary:**

This paper studies the continuum-armed bandit problem, which is an extension of the traditional multi-armed bandit. Specifically, the authors propose an explicit representation using an orthogonal feature map (e.g. based on Fourier, Legendre functions) to transform the original problem into a linear bandit with misspecification. And the authors develop two algorithms named OB-LinUCB and OB-PE and use a suite of simulations to verify the efficiency of the proposed algorithms.

**Strengths:**

Most parts of the paper are quite clear and easy to follow.

Based on my knowledge it is new to use orthogonal function bases to transform the continuum-armed bandit into misspecifed linear bandits. The simulations also showcase the high efficiency of proposed algorithms.

I haven't checked all details of the proof in the Appendix, but I feel they should be correct. I will refer to other reviewers' opinions as well.

**Weaknesses:**

1. Nowadays, the study on continuum-armed bandit also focuses on the general metric space. (e.g. Zooming algorithm mentioned in your work studies arbitrary space with any distance). Could you extend your algorithm to general metric space? I feel it should be possible if we can construct a decent orthogonal basis on any metric space, is that true? But how to construct the function base is required but unknown.

2. I agree it is relatively hard to explore multidimensional space $[0,1]^d$, but I feel the current results in this work on multidimensional space are still weak. From Table 1 the proposed OB-PE can achieve non-optimal regret bound under the condition $d < 2s$, which is not up to the state-of-the-art literature in this area. For implementation, both the number of arms and the dimension of arms would increase exponentially, and hence I am not sure whether it can still perform well in multidimensional space.

3. Without knowing the value of $T$, I am not sure if the proposed algorithm still works since some settings (e.g. $N$) rely on the value of $T$.

4. For the presentation of Algorithm 1, why don't the authors specify how to discretize the arm space directly there? It seems that discretization is unavoidable for OB-LinUCB and OB-PE.

**Questions:**

Besides my concerns in the above Weaknesses section, could you also answer my question about your experiment as follows:

1. It seems that you only use OB-LinUCB which is lack of strong theoretical support in your experiment. Why don't you use OB-PE as well since it has some good theoretical property instead? It seems there is some inconsistency between theory and experiments.

2. I may overlook, but how many arms do you choose in experiments for OB-LinUCB? I think you discretize the interval [0,1] evenly (maybe with $\sqrt{T}$ arms). Do you have any theoretical support for OB-LinUCB when discretizing the arm set? I think this is very important and should be illustrated in your main paper clearly.

3. Some experiments one high-dim space (e.g. [0,1]^2) will make your experiments more reliable. I am concerned on the computational issue of the algorithm since the number of arms and dimension may explode exponentially.

---

> ### Author Response · Authors · 2023-11-15
> **Answer to the reviewer**
>
> **Continuum-armed bandit on the general metric space.** Unfortunately, generalizing this approach to that setting would be highly nontrivial, as polynomial and trigonometric features are not even defined on arbitrary metric spaces. On the other side, the main competitor of this algorithm, UCB-meta-algorithm, suffers the same issue, also because the concept of "differentiability" itself cannot be trivially defined.
>
> **Computational performance in multidimensional space.** Please see the common answer about computational complexity.
>
> **Does the algorithm work without knowing the value of $T$?** Please see the common answer about adaptivity.
>
> **For the presentation of Algorithm 1, why don't the authors specify how to discretize the arm space directly there?** In fact, discretization is unavoidable for all the algorithms considered in the state of the art, the only exception being Zooming, which without discretization is NP-Hard. Therefore, to grant a fair comparison, we preferred not to specify the discretization to show in Table 1 how the performance of every algorithm depends on the number of its elements. Indeed, it is true that in the general case, a uniform $T^{-1/2}-$cover would be the best choice, but also emphasizing the dependence on the cardinality $k$ of a general discretization is important.
>
> **Use of OB-LinUCB rather than OB-PE in the experiments** We made this choice based on the principle that, usually, algorithms that optimize the worst-case regret are not the best option in practice. Coherently with this choice, we have compared our algorithm not with BPE, which achieves the best regret in kernelized bandits, but with IGP-UCB, which is well-known to have superior performance in practice regardless of the worse regret bound. Still, since also the practical performance of OB-PE is an interesting question, we are going to perform the experiments with this algorithm and upload them in a new version of the paper before the end of the discussion period.
>
> **Discretization in the experiments** As the reviewer has understood, we discretize the interval [0,1] evenly with $\sqrt T$ arms. Unfortunately, OB-LinUCB is not endowed with a regret bound for this setting, and it is only shown as the most natural algorithm that can be built with orthogonal function representation.
>
> **Some experiments one high-dim space (e.g. $[0,1]^2$) will make your experiments more reliable. I am concerned on the computational issue of the algorithm since the number of arms and dimension may explode exponentially.** First point: Being the algorithm valid for $d>1$, it is indeed a good idea to add an experiment in case of a bi-dimensional environment. We will do it and upload it in a new version of the paper before the end of the discussion period. Unfortunately, it will not be possible to include also the baselines used in the current experiment, as their running time is too high, and we cannot have the results in time. Second point: See the common answer about computational complexity.
>
> We thank the reviewer for their comments and have hope to have addressed all their questions.

---

> > ### Comment · Reviewer_DACT · 2023-11-22
> > **Thank you for your responses and additional experiments**
> >
> > I'd like to appreciate authors' responses to my questions and the additional experiments made for OB-PE. I believe this work has its merit and some extensions such as to general metric space is highly non-trivial. And the idea of introducing orthogonal feature mapping is new and interesting. However, I still have concern on some presentation of the work: I feel it would be better to mention how to discretize the space explicitly in the pseudocode in detail. And the computational benefit of the algorithm is not very general. Therefore, I feel this paper is right on the borderline, and I will keep my score for now.

---

### Official Review · Reviewer_MY3R · 2023-10-26

**Soundness:** 4 excellent
**Presentation:** 3 good
**Contribution:** 3 good
**Rating:** 6
**Confidence:** 3

**Summary:**

The paper resolves the continuous arm bandits where the reward function is smooth. By introducing orthogonal Fourier and Legendre feature maps, the paper shows that an approximation of the smooth function is possible even when the feature map is not available to the learner. The proposed algorithm achieves the nearly optimal regret bound in $d=1$ dimensional cases and in $d>1$ dimensional cases where the reward function is analytic. Empirical results show fair performance of the algorithm with computational efficiency.

**Strengths:**

The paper eliminates the assumptions on the reward functions in bandit problems with continuous arms and without Gaussianity, by constructing an orthogonal feature map. The proposed algorithm is principled and computationally efficient for estimating general reward functions. Helpful discussion on the hardness of deriving optimal regret bound on multidimensional arms navigates the future work direction.

**Weaknesses:**

(a) In Theorems 3-5, the choice of $N$ requires the knowledge of $T$.
(b) There is a large gap between the choice of $N$ in theoretical results and empirical results. The performance and the computational time of the proposed algorithms seem to be heavily affected by the choice of $N$. Numerical results and discussions on different choices of $N$ seem missing.

**Questions:**

Q. Could the algorithm be modified to execute without knowing $T$ a priori?
Q. How could we choose $N$ when we do not know the true reward function? If we choose $N$ as in Theorems 3-5, would computation be heavy?
Q. How does the choice of $N$ affect the performance and computational time of Fourier UCB and Legendre UCB?
Q. In Figures 1 (b) and 2 (b), why do Fourier UCB and Legendre UCB perform worse when they use only even functions and the true reward function is even?

---

> ### Author Response · Authors · 2023-11-15
> **Answer to the reviewer**
>
> **Adaptivity for unknown $T$ and how to choose $N$ without $s$.** See the common answer about adaptivity.
>
> **Choice of $N$ in the experiments.**
> Keeping the values of $N$ in the experiments disconnected from theory was a deliberate choice to show that the requirement to know the differentiability of the reward function exists only in theory. In fact, having chosen a fixed value of $N$ for functions with very different levels of regularity suggests that this parameter can be chosen in a totally empirical way and tuned very easily.
>
> **Computational complexity dependence on $N$.**
> This question is linked to the theme of computational complexity, therefore, we suggest the reviewer to also see the "Computational complexity" section in the answer to all reviewers. In short, the computational complexity scales as $N^{3d}$, which means $\widetilde N^3$, where $\widetilde N$ is the number of features. This is unavoilable while using algorithms for linear bandits, as inversion of a design matrix is necessary. Still, note that the algorithm chooses $N$ in a way that $N^{d}\le T$, so that the full computational complexity does not explode.
>
> **In Figures 1 (b) and 2 (b), why do Fourier UCB and Legendre UCB perform worse when they use only even functions and the true reward function is even?**
> We thank the reviewer for noticing this subtle phenomenon. The reason stays in the true reward function: as it is a polynomial of degree 4, including in the feature map even polynomials of higher degree is as useless as including odd functions. In fact, as high-frequencies are more prone to overfitting the noise, the effect of including these features may be even worse than adding other useless components, as in this example.
>
> We thank the reviewer for their comments and insightful observations and hope to have addressed all their questions.

---

> > ### Comment · Reviewer_MY3R · 2023-11-22
> >
> > Thank you for the response. It addresses most of my questions. But still, I have some questions on $N$
> > Q. Could you explain why $N$ can be tuned very easily?
> > Q. If the algorithm chooses in a way $N^d \le T$, is the condition $N=T^{\frac{d}{2s}} violated? If $N$ is chosen empirically, does the theoretical regret guarantee still hold?

---

> > > ### Author Response · Authors · 2023-11-22
> > > **The role of $N$**
> > >
> > > Indeed, the fact that $N$ is tuned very easily is only empirical, and in case of using an $N$ different from the one with theoretical guarantees the only possible regret bounds are analogous to the ones obtained for a misspecification of $s$ discussed previously.
> > >
> > > In fact, there is a phenomenon similar to what happens with Fourier series: despite the theory telling us that Fourier series converge only under very specific hypotheses, their approximation power in practice is universally recognized. The reason is that the counterexamples of functions for which Fourier series converge slowly, or do not converge at all, are often “pathological” functions with characteristics that are never observed in reality, given that the functions used in real problems are usually piecewise regular. This phoenomenon also applies to Legendre polynomials, as we observe in the experiments : since
> > > 1. what really matters in continuous armed bandit algorithms is to estimate the function near the maximum
> > > 2. the convergence of Fourier series/Legendre polynomials for piecewise regular functions is very rapid (unless one is near a point of discontinuity)
> > >
> > > excellent results can be obtained even for values of $N$ much smaller than those for which there are theoretical guarantees. This means that by taking a reasonably small value of $N$, our algorithm exhibits excellent results on regret with exceptional computational complexity.

---

### Official Review · Reviewer_hB5r · 2023-11-01

**Soundness:** 3 good
**Presentation:** 3 good
**Contribution:** 2 fair
**Rating:** 5
**Confidence:** 3

**Summary:**

This paper studies continuum arm bandits with a generic smoothness assumption on the reward being $s$ times continuously differentiable. The main new idea of the paper is that by using a finite representation of the reward function in terms of a known orthogonal basis for the function class, one can reduce the problem to misspecified linear bandits. This leads to an algorithm with optimal regret in some regimes and better computation time than prior algorithms in the literature.

While I think the idea of this paper is novel, the theoretical regret bounds are only optimal in some regimes and the overall benefit especially over the cited algorithm UCB-Meta-algorithm is unclear.

**Strengths:**

* The paper is easy to read and well-written despite being quite math heavy, and I found the main algorithmic idea straightforward to follow.
* The thorough comparison with prior works in Section 5.3 was helpful for placing this result in context with the rest of the literature.

**Weaknesses:**

* It seems like the theoretical regret upper bounds require tuning $N$ with knowledge of the underlying reward function's smoothness $s$. This seems like an unrealistic assumption to me. Can the authors comment on misspecified $s$ or adapting to unknown $s$?
* In order to even use the orthogonal features (at least for dimension $d=1$), it seems like one also needs $(s+1)$-th derivative of $f$ to be square integrable. Is this a reasonable assumption to make for common reward functions in stochastic optimization? Some discussion on this would be nice.
* Looking at Table 1, it seems like the UCB-Meta-algorithm attains the optimal regret, while the main procedure of this paper OB-PE does not get optimal regret for $d>1$. The paper claims UCB-Meta-Algorithm has no regret guarantee for infinitely-differentiable rewards, but I don't see why I can't just run UCB-Meta-Algorithm with very large Holder exponent $s \gg d$ which should get a regret bound which is nearly $T^{1/2}$ acccording to bounds of Liu et al., 2021. Thus, the only real advantage of OB-PE seems to be in time complexity.
* There is a discrepancy between experiments and theory in the sense that the paper uses two different algorithms OB-PE vs OB-LinUCB for theoretical regret bounds vs experiments. It's not explained why OB-PE is not implemented or analyzed in experiments. Some explanation on this would be nice.

**Questions:**

# Questions
* See above in "Weaknesses".
* What is the dependence of the regret bounds of OB-PE on the Lipschitz constant $L$? In Lipschitz bandits literature, this is well known to be $L^{\frac{d}{d+2}}$, but it is not clear to me what dependence appears here.
* Theorem 4 for the $d>1$ dimensional regret upper bound does not seem to have any condition on the $(s+1)$-th partials of $f$, which seems wrong to me since Theorem 3 required it. Why is this?

# Writing Suggestions/Typos
* The regret formula in the first paragraph should have the terms reversed in the difference.
* In the fifth paragraph of page 1, the domain $[a,b]$ of $\phi$ should be $[a,b]^d$?
* I was confused why initially the reward function $f:\mathcal{X} \to \mathbb{R}$ has unbounded scale, yet all the regret bounds seem to be scale-free. This is because the paper later on assumes $\|f\|_{\infty}=1$, i.e. assumes knowledge of the scale of $f$. It might be better to just define bounded reward $f:\mathcal{X}\to [0,1]$ from the outset to not mislead readers.
* Many environment references throughout the writing (e.g., algorithm 1, appendix 1, equation (1)) should have their names capitalized (e.g., Algorithm 1).
* In Section 4.2, the Lipschitz constant $L$ is used before it is defined.
* In the writing, "s+1-th derivative" would read better as "(s+1)-th derivative".

---

> ### Author Response · Authors · 2023-11-15
> **Answer to the reviewer**
>
> **Adaptivity on $s$.** See common answer about adaptivity.
>
> **Assuming square-integrable derivative in practice.** The setting of stochastic bandits has achieved relevant success in the field of pricing. In this area, bandit algorithms are used to optimize a function of the price $p$ given by $f(p):=p d(p)$ where $d(p)$ is an unknown function known as "demand curve", which is non-increasing in $p$ (ref. "Pricing the long tail by explainable product aggregation and monotonic bandits" by Mussi et al.). Of course, the differentiability of $f(\cdot)$ corresponds to the one of $d(\cdot)$, the demand curve, and since this function is decreasing and bounded in $[0,1]$, its derivative cannot be "too large too often". This can be formulated by asking that its derivative is square-integrable.
>
> **Infinitely differentiable functions.** We agree with the reviewer that, being $\mathcal C^\infty$ included in $\mathcal C^s$ for every $s>0$, the algorithm from Liu et al., 2021 can achieve regret of order $\mathcal O(T^{1/2+\beta})$ for every $\beta > 0$ when the function is infinitely differentiable. The point is that UCB-Meta-Algorithm has to choose a number of features that depends on $\beta$ and diverges for $\beta \to 0$ (that algorithm relies on dividing the actions space in hypercubes and running a linear bandit algorithm in each hypercube. We refer to the number of features used by each of these linear bandits). Conversely, our algorithm uses fewer features the higher the value of $s$, and in particular, for $s=+\infty$, it can achieve regret of order $\sqrt T$ using the lowest number of features. This strange phenomenon makes the two algorithms complementary, with UCB-Meta-Algorithm being more suitable for large $d$ and small $s$, and our one for $d=1$ or $s=+\infty$.
>
> **Use of OB-LinUCB in the experiments.**  We made this choice based on the principle that, usually, algorithms that optimize the worst-case regret are not the best option in practice, where simplicity works much better. Coherently with this choice, we have compared our algorithm not with BPE, which achieves the best regret in kernelized bandits, but with IGP-UCB, which is well-known to have superior performance in practice regardless of the worse regret bound. In the end, if ever one of our algorithms has a hope to be used in practical scenarios, this is probably OB-LinUCB, which is the most natural marriage between a Linear Bandit algorithm and the orthogonal function representations. Still, we will implement OB-PE and let the reviewer have its result by the deadline of the discussion period.
>
> **Dependence of the regret bounds of OB-PE on the Lipschitz constant $L$.** Thanks for the question, which opens up some reflections we hadn't considered. Our regret bound is roughly of order
>     $$R_T \le \mathcal O\left (\sqrt{N^d T}+LN^{d/2}N^{-s}T\right ),$$
> where $L$ is the Lipschitz constant of the $s-1$ derivative. With the optimal choice $N=L^{1/s}T^{1/2s}$ we get, in the regime $d<2s$ where we have regret guarantees,
>     $$R_T \le \mathcal O\left (L^{d/(2s)}T^{(2s+d)/(4s)}\right ),$$
> meaning that the dependence is $L^{d/(2s)}$ (which of course becomes vacuous for $d>2s$, when also the regret is unbounded).
>
> **Assumptions of Theorem 4.** The different type of assumption reflects the difference in the type of analysis behind the result. In fact, given that the theory of Legendre polynomials has so far been developed only in one dimension, it is not possible to use it in the general case, and the argument used there does not make assumptions on the $(s+1)$-th derivative. This is linked to the reason why our analysis is sub-optimal for $d>1$.
>
> We thank the reviewer for their comments and for identifying some typos in our paper. We hope we have addressed all of their concerns.

---

> > ### Comment · Reviewer_hB5r · 2023-11-20
> >
> > Thank you for the clarifying response. While I originally believed the main benefit of the proposed algorithm was in reduced computational complexity, reading the other discussions and general rebuttal, it seems like even that comparison is only superior in certain regimes of $d,s$. Overall, I think the "reduction" of infinite-armed bandits to linear bandits is interesting, but the overall benefit still seems to be lacking. As such, I will keep my score the same.

---

### Official Review · Reviewer_hchS · 2023-11-02

**Soundness:** 2 fair
**Presentation:** 2 fair
**Contribution:** 2 fair
**Rating:** 5
**Confidence:** 3

**Summary:**

Consider a continuous arm bandit problem where the action space is $[0,1]^d$. While Reproducing Kernel Hilbert Space (RKHS) feature maps have been used previously, they present computational difficulties. The authors propose an approach that uses an orthogonal feature map to convert the problem into a linear bandit problem. This approach not only offers competitive regret guarantees compared to existing methods, but also reduces computational complexity.

**Strengths:**

An analysis of linear bandit algorithms under misspecification is utilized for continuous armed bandits, proposing an algorithm with a small computational complexity.

It demonstrates competitive performance numerically as well.

**Weaknesses:**

A significant portion of this paper seems to be dedicated to presenting classical results. While understanding the foundation is essential, I would encourage the authors to consider focusing more on their unique contributions to the field.

From my understanding, the proposed algorithm/analysis appears to present a solution for an efficient trade-off between misspecification (bias) and regret (variance) in the continuous arm bandit problem. However, the contribution seems to lie primarily in combining these elements, and I did not find the approach significantly novel. If there is technical novelty, it should be better presented and emphasized.
The paper does not clearly address the potential for the $d/s$ value to become significantly large (though not infinite). In such cases, the computational advantages of the proposed method compared to existing methods may not only be positive but could potentially be negative. This issue needs to be addressed.

The algorithm seems to require a substantial amount of inputs (values dependent on $T, N, s$, the choice of basis, etc.). How much does this limit its adaptability? Are there methods that are agnostic to $T$ or $s$? Or is this level of input requirement comparable to existing research? These questions are crucial when discussing the adaptability of the algorithm.

Considering the high standards of ICLR, it is unclear whether the contributions of this paper are significant enough to warrant acceptance. The authors might want to better highlight the novelty and impact of their work.

**Questions:**

As I also wrote in the previous section:

- What is the technical novelty of the proposed algorithm/analysis? How is it emphasized?
- Can you explain how the proposed method addresses the potential for the d/s value to become significantly large?
- To what extent does the substantial number of inputs required by the algorithm limit its adaptability?
- Are there methods that are agnostic to T or s?
- How does the level of input requirement in this paper compare to existing research?

---

> ### Author Response · Authors · 2023-11-15
> **Answer to the reviewer**
>
> **Paper organization**
> The theory behind our algorithm is highly nontrivial for someone with a background different than mathematics. Therefore, we have decided to keep in the main paper the parts that are essential for comprehending the paper and leave the more advanced and subtle discussions in the appendix.
>
> **Novelty**
> We acknowledge the reviewer's observation that our analysis relies heavily on theorems already established in the field of functional analysis. However, we disagree with the notion that this diminishes the novelty of our work. On the contrary, given that existing literature predominantly employs Kernel methods or focuses on finding a cover for the arm space, the utilization of a theory previously unexplored in this context should be regarded as a valuable contribution. Recent research in bandits on Reproducing Kernel Hilbert Spaces (RKHS) has revealed that performance guarantees are intricately tied to the decay properties of the eigenvalues of the kernel. In our study, we demonstrate (refer also to appendix E.2) that the same holds true for general continuous armed bandits. This suggests that, even if certain results in our paper fall short of optimality, the methods formulated here could be applied to extend many findings from the extensive RKHS theory to this specific setting.
>
> **Computational advantages where the d/s value to become significantly large** See the common answer about computational complexity.
>
> **Adaptivity for values of $s$ and $T$** See the common answer about adaptivity.
>
> **Input requirement in this paper compared to existing research** All the related papers, like ours, assume knowledge of both the time horizon $T$ and the order $s$ of differentiability. In particular, the four works on Bayesian optimization also presume working in a Reproducing Kernel Hilbert Space (RKHS) with a _known_ kernel. Since the knowledge of the kernel directly determines the order of smoothness of the functions considered in the RKHS, their requirement is notably more stringent than ours.
>
> We thank the reviewer for their comments and hope to have addressed all their questions.

---

> > ### Comment · Reviewer_hchS · 2023-11-22
> > **Thank you for the rebuttal**
> >
> > Thank you for your response, but I'm still not clear on why it's novel to combine the results of the misspecification bandit and the functional analysis. If it's simply a matter of combining the results, it seems far from the standards of ICLR. Were there any challenges that arose from combining the existing results? If so, what were the steps you took to overcome those challenges?

---

> > > ### Author Response · Authors · 2023-11-22
> > > **Novelty**
> > >
> > > Since our article is presenting a new line of research that transcends the two strands to which all the remaining literature on continuous armed bandit belongs, we are perplexed by the fact that the novelty is being contested. If the point pf the reviewer is about the technical difficulty of the proofs, here we have a non-exaustive list of the challeges that we faced:
> > >
> > > 1. Using a standard linear bandit algorithms leads to a regret bound of the form $\tilde N\sqrt{T}$, while to achieve the regret bound we needed $\sqrt{\tilde NT}$. This problem was solved by using a particular algorithm for linear bandits which was very recently invented, and an argument which is explained in the main proof.
> > > 2. For practical bandit algorithms, other polynomial bases are often used to do representation learning, such as Bernstein polynomials. Instead, our argument uses the orthogonality property of Legendre polynomials, which leads to a form which is good for linear bandit algorithms trough the use of Parseval's theorem.
> > > 3. The proof in the three cases ($d=1, s<+\infty, s=\infty$) are all different since there are holes in the mathematical theory that do not allow to use the same proof strategy in the three cases. In particular, the last case, which is not explicitly faced in the rest of the literature, has proof which do not relate with pre-existing results.
> > >
> > > Still, is this the real aim of a Machine Learning paper? The core of our work is to propose a new approach based on a kind of theory that was never used in the Bandit setting. Incorporating new results from contemporary mathematics to create new algorithms should be considered something valuable: even [1], one of the most celebrated papers of the whole bandit literature, is substantially built on the idea of applying self normalized processes [2] to linear bandits.
> > >
> > > In the end, it may seem that our paper lacks novelty since the proposed algorithm is a simple modification of existing ones. However, we think that, on the contrary, proposing a simple solution that exploit known mathematics to the fullest is preferable than re-inventing the wheel. In the end, quoting someone that was quite recognised to his novelty, simplicity is the ultimate sophistication.
> > >
> > > [1]  Y Abbasi-Yadkori, C Szepesvári, D Pál, "Improved algorithms for linear stochastic bandits"
> > >
> > > [2] Victor H de la Pena, Tze Leung Lai, Qi-Man Shao, "Self-normalized processes: Limit theory and Statistical Applications"

---

### Author Response · Authors · 2023-11-15
**Rebuttal: Common points**

Since some points of paramount importance were raised by multiple reviewers, we decided to give a common answer in this section.

**Computational complexity** is a crucial issue in this kind of machine learning application, and one of the most important features of our work is the limited complexity of our algorithm. While for $d=1$ the result of the experiments shows unmistakable evidence that the running time is enormously reduced with respect to the state of the art, questions have been posed over computational complexity for $d>1$.

1. **Discretization**. Although it is true that the optimal discretization has roughly $T^{d/2}$ elements, a number exploding with $d$, this is an unavoidable consequence of the curse of dimensionality. Every algorithm in state of the art requires to pass through a similar procedure: UCB-Meta-algorithm needs to discretize both the action space into bins and then discretize each bin to solve the linear bandit problem; the four Bayesian Optimization baselines require discretizing the action set to find the maximum of the posterior; Zooming needs to start from a discretization too, otherwise it definition implies solving an NP-Hard problem. Table 1 contains the computational complexity of every algorithm with respect to both $T$ and the cardinality $k$ of the discretization, showing that our algorithm only scales as $T^{1/2}k$ in the worst case, while for the others, at least in $Tk$.

2. **Dimension of the linear bandit problem**. In multidimensional spaces, the dependence between the order of approximation $N$ and the dimension $\widetilde N$ of the auxiliary linear bandit problem built by OB-PE and OB-LinUCB is $\widetilde N = N^d$. Nevertheless, with the optimal choice is $N=\mathcal O(T^{\frac{1}{2s}})$, this does not exceed $\widetilde N = T^{\frac{d}{2s}} \le T$ in the feasible regime $d<2s$. Considering that the computational complexity scales as $\widetilde N^3$, this term is bounded by $T^{\frac{3d}{2s}}$, as reported in Table 1.

**Adaptivity** is a very relevant feature of every machine learning algorithm. In our setting, it is interesting to see if our algorithm is able to perform well in case information about either the time horizon $T$ or the order of smoothness $s$ is missing.

1. Adaptivity with respect to $T$. Even if our algorithm requires the value of $T$ to compute the number $N$ of features, we can achieve a similar regret guarantee, except for a factor $\log(T)$ by first guessing a small value $T_0$ for $T$, running the algorithm for horizon $T_0$ and then continuing doubling the time horizon ($T_0, 2T_0, 4T_0, \dots$) until $T$ is reached. In this way, the algorithm performs similarly, being agnostic of $T$ (see [1]).

2. Adaptivity with respect to $s$. The other relevant parameter which may be unknown to the agent is $s$.
        Let us assume the algorithm is run with $s_{\text{miss}} \le s_{\text{true}}$. In this case, since $\mathcal C^{s_{\text{miss}}}(\Omega)\subset \mathcal C^{s_{\text{true}}}(\Omega)$, we have the regret bound corresponding to the lower smoothness parameter $s_{\text{miss}}$, so that
        $$R_T \le \mathcal O\left (T^{\frac{2s_{\text{miss}}+d}{4s_{\text{miss}}}}\right).$$

        If instead the algorithm is run with $s_{\text{miss}} > s_{\text{true}}$, we have

        $$R_T \le \mathcal O\left (T^{\frac{4s_{\text{miss}}-2s_{\text{true}}+d}{4s_{\text{miss}}}}\right);$$

        indeed, using a misspecified value $s_{\text{miss}}$ leads to a misspecified value for $N=\mathcal O(T^{\frac{1}{2s_{\text{miss}}}})$. Substituting this value into the regret bound depending on $N$, which is of order
        $$R_T \le \mathcal O\left (\sqrt{N^d T}+N^{d/2}N^{-s}T\right ),$$

        we get exactly the previous bound. Note that in the experiments, $s$ **is already unknown**. In fact, we run the algorithm on all the environments with the same value for $N$ to show that choosing $N=\mathcal O(T^{\frac{1}{2s}})$ is not crucial in practice.

[1] Besson, L., & Kaufmann, E. (2018). What doubling tricks can and can't do for multi-armed bandits. arXiv preprint arXiv:1803.06971.

---

> ### Author Response · Authors · 2023-11-21
> **Updated paper**
>
> We have incorporated the experiments suggested by the reviewers into the paper. The results of these experiments are available in the appendix, F.4. Specifically, we have added experiments with OB-PE on the environments of the paper, as well as an experiment in dimension 2.

---

### Meta-Review · Area_Chair_2xrW · 2023-12-05

**Metareview:**

This is a borderline paper with one major criticism being that the approach is incremental and lacks novelty. Reading through the rebuttal I agree with the reviewers. For example, the authors highlight that performance guarantees are intricately tied to the decay properties of the eigenvalues of the kernel function. But this is long known in the RKHS literature in other contexts and minimax optimal rates are attained in settings like ridge regression by utilizing the eigenvalue sequences. It is not surprising at all that the very same properties also factor in in the bandit context.

**Justification For Why Not Higher Score:**

It is a borderline paper and a weak accept is also an option. The reject is mainly based on the lack of novelty and the incremental character of the paper.

**Justification For Why Not Lower Score:**

n/a

---

### Decision · Program_Chairs · 2024-01-16

Reject